# Refining data-data and data-model vegetation comparisons using the Earth Movers' Distance (EMD)

Manuel Chevalier[1,2], Anne Dallmeyer[3], Nils Weitzel[4,5], Chenzhi Li[6,7], Jean-Philippe Baudouin[4,5], Ulrike Herzschuh[6,7,8], Xianyong Cao[9], Andreas Hense[1]

5     1 Institute of Geosciences, Sect. Meteorology, Rheinische Friedrich-Wilhelms-Universität Bonn, Germany
      2 Institute of Earth Surface Dynamics, Géopolis, University of Lausanne, Switzerland
      3 Max Planck Institute for Meteorology, Bundesstrasse 53, 20146 Hamburg, Germany
      4 Department of Geosciences, University of Tübingen, Schnarrenbergstr. 94-96, 72076 Tübingen, Germany
      5 Institute of Environmental Physics, Heidelberg University, Im Neuenheimer Feld 229, 69120 Heidelberg, Germany
10    6 Polar Terrestrial Environmental Systems, Alfred Wegener Institute Helmholtz Centre for Polar and Marine Research, Telegrafenberg A45, 14473 Potsdam, Germany
      7 Institute of Environmental Science and Geography, University of Potsdam, Karl-Liebknecht-Str. 24–25, 14476 Potsdam, Germany
      8 Institute of Biochemistry and Biology, University of Potsdam, Karl-Liebknecht-Str. 24–25, 14476 Potsdam, Germany
15    9 Alpine Paleoecology and Human Adaptation Group (ALPHA), State Key Laboratory of Tibetan Plateau Earth System, Resources and Environment (TPESRE), Institute of Tibetan Plateau Research, Chinese Academy of Sciences, 100101 Beijing, China

*Correspondence to*: Manuel Chevalier (mchevali@uni-bonn.de)

**Abstract.** Comparing temporal and spatial vegetation changes between reconstructions or between reconstructions and model simulations requires carefully selecting an appropriate evaluation metric. A common way of comparing reconstructed and simulated vegetation changes involves measuring the agreement between pollen- or model-derived unary vegetation estimates, such as the biome or plant functional type (PFT) with the highest affinity scores. While this approach based on summarising the vegetation signal into unary vegetation estimates performs well in general, it overlooks the details of the underlying vegetation structure. However, this underlying data structure can influence conclusions since minor variations in pollen percentages modify which biome or PFT has the highest affinity score (*i.e.* modify the unary vegetation estimate). To overcome this limitation, we propose using the Earth Movers' Distance (EMD) to quantify the mismatch between vegetation distributions such as biome or PFT affinity scores. The EMD circumvents the issue of summarising the data into unary biome or PFT estimates by considering the entire range of biome or PFT affinity scores to calculate a distance between the compared entities. In addition, each type of mismatch can be given a specific weight to account for case-specific ecological distances or, said differently, to account for the fact that reconstructing a temperate forest instead of a boreal forest is ecologically more coherent than reconstructing a temperate forest instead of a desert. We also introduce two EMD-based statistical tests that determine 1) if the similarity of two samples is significantly better than a random association given a particular context and 2) if the pairing between two datasets is better than might be expected by chance. To illustrate the potential and the advantages of the EMD

and the tests in vegetation comparison studies, we reproduce different case studies based on previously published simulated and reconstructed biome changes for Europe and capitalise on the advantages of the EMD to refine the interpretations of past vegetation changes by highlighting that flickering unary estimates, which gives an impression of high vegetation instability, can correspond to gradual vegetation changes with low EMD values between contiguous samples (case study 1). We also reproduce data-model comparisons for five specific time slices to identify those that are statistically more robust than a random agreement while accounting for the underlying vegetation structure of each pollen sample (case study 2). The EMD and the statistical tests are included in the paleotools R package (https://github.com/mchevalier2/paleotools).

## 1 Introduction

Fossil pollen records are commonly used to evaluate Earth System Model (ESM) palaeosimulations in the climate space (*i.e.* pollen data are converted into climate parameters using transfer functions, Birks et al., 2010; Chevalier et al., 2020) and the vegetation space (*i.e.* vegetation features are simulated using vegetation models, *e.g.* Prentice et al. (1998), Tian et al. (2018), Wohlfahrt et al. (2008)). Both evaluations are necessary to explore the strengths and weaknesses of fossil pollen data, climate and vegetation models, and the modern observations that link vegetation with climate. To compare data and models in the vegetation space, the pollen data are commonly translated into plant functional types (PFTs) – which are defined by the plant species' life forms, leaf forms, phenologies, and bioclimatic tolerances and reflect their adaptations to environments (Prentice et al., 1996, 2000; Prentice and Webb III, 1998) – or biomes (macro-ecosystems) – which correspond to broad vegetation classification units characteristic of regional- to global-scale features (e.g. Cao et al. (2019), Dallmeyer et al. (2017), Sato et al. (2021)). This transformation is performed by calculating an affinity score that measures the similarity of the pollen sample with the studied PFTs or biomes for each pollen sample based on a set of predefined rules.

Transforming raw pollen percentages into biome or PFT affinity score distributions has several advantages, as (*i*) it reduces the dimensionality of the vegetation space (*i.e.* reducing the few hundred pollen taxa usually observed across a continent to about 10-30 biomes or PFTs), (*ii*) it summarises the main traits characterising the studied vegetation compositions (*i.e.* enabling a convergence of the traits and a spatial homogenisation of the data), and (*iii*) it improves the comparability of data with different origins (*i.e.* pollen data, modern observations, and simulations). The PFT or biome with the highest affinity score is ultimately labelled the most representative (Prentice et al., 1996, 2000). The transformed pollen data can thus be directly compared with model simulations of the same period (e.g. Cao et al., 2019; Prentice et al., 1998) or other pollen data of different periods (*e.g.* Allen et al., 2020) and the "agreeing" and "disagreeing" pairings (*i.e.* binary assessments of the compared biome or PFT estimates with the highest affinity score) are counted to determine the global similarity of the compared datasets.

However, simplifying affinity scores to a unary biome or PFT estimate can overly homogenise the data. When the highest affinity score is much larger than the second-highest score, reducing the affinity score distributions to one PFT or biome is a

reasonable simplification. In contrast, when the difference between the highest and second-highest affinity score is minor, simplifying multidimensional data to one unary estimate leads to ignoring a significant part of the information conveyed by the affinity score distributions as a representative fraction of the fine-scale details of the vegetation structure gets lost in such situations. In particular, this suggests that many distinct affinity score distributions can lead to the same PFT or biome with the highest score. In addition, this simplification disregards the natural uncertainties of biome estimates from pollen samples that arise from, for instance, varying pollen productivity of taxa, limited taxonomic resolution or long-distance pollen transport.

This issue is further illustrated by the two samples shown in Fig. 1A based on the biomised data of Cao et al. (2019). The 'Evergreen taiga' biome has the highest score in both samples, rendering them indistinguishable when reduced to their biome with the highest affinity score. However, inspecting their affinity score distributions informs us about significant differences, where the top sample most likely represents a well-forested environment, while the bottom sample is closer to a mosaic of forest patches connected by open landscapes (characterised by more abundant Tundra and Grassland pollen taxa). In contrast, the two biome affinity score distributions in Fig. 1B have distinct biomes with the highest affinity scores. They are thus classified into different vegetation groups (one is classified as Evergreen Taiga and the other as Cool/Cold forest), even if their affinity score distributions only differ by minute changes between the different forest biomes. In such cases, the two distributions likely represent (very) similar environments despite having their highest affinity with different biomes.

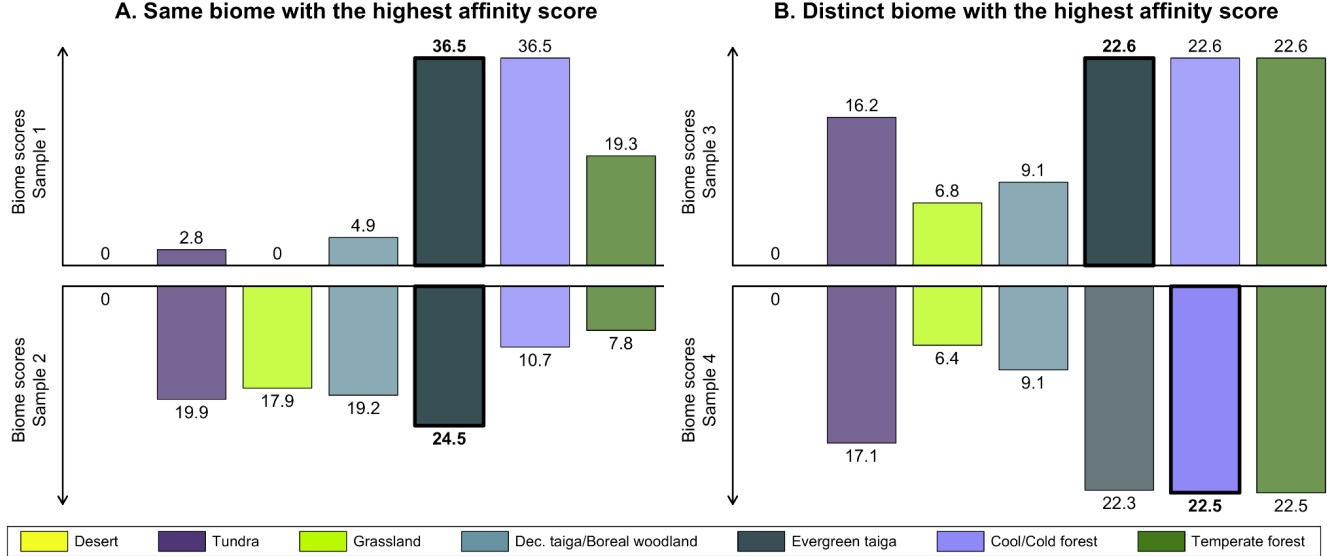

**Figure 1. Illustration of the limitation of using unary biome estimates to compare samples. (A)** In the two samples, the biome with the highest affinity score is the same (Evergreen taiga), while the affinity score distributions have notable differences. **(B)** The biomes with the highest affinity score are different (Evergreen taiga and Cool/Cold forest), but their affinity score distributions only differ by minor differences. The data are reproduced from the study of Cao et al. (2019).

The approach of summarising an array of affinity scores by its PFT or biome with the highest score is thus insufficiently sensitive, as it can lead to a loss of accuracy when comparing datasets (contrasting samples can be assigned to the same category, while similar samples can be assigned to different categories; Fig. 1). Employing continuous metrics that consider the entire affinity score distributions (as opposed to comparisons of their PFT or biome with the highest score) can thus refine the quality of data-data and data-model comparisons. Many distances commonly used to compare pollen data – such as the Manhattan/Euclidean distance (*i.e.* calculating the absolute/squared differences between the scores of the same biomes) or the squared-chord distance (e.g. Overpeck et al., 1985) – could be used to measure the dissimilarity of two affinity score distributions. Pollen-based biome scores have also been compared to biome scores estimated from the net primary productivity per PFT produced by LPJ-GUESS using canonical correlation analyses (Allen et al., 2010). In all cases, these metrics give the same importance to all the differences without accounting for the fact that all vegetation changes are not ecologically equivalent. For example, replacing a cool/cold forest with a temperate forest represents a more minor ecological/climatic shift than replacing a forest with a desert, even if the absolute differences in biome scores are the same (*e.g.* Allen et al., 2020). To account for these two limitations, we propose the Earth Movers' Distance (EMD) metric as a new way to compare pollen-derived affinity score distributions with one another and with vegetation simulations. The advantage of this distance metric compared to standard binary assessments based on unary estimates is dual: 1) the EMD is a continuous metric such that vegetation differences can be quantified in finer detail, and 2) the inclusion of ecologically-informed weights adds a level of refinement that takes into account different types of mismatches between samples.

This paper first introduces the EMD and describes the properties that make it well-suited to capitalise on the information contained in affinity score distributions. Then, using a series of illustrative case studies based on the already-published biomised data and simulations of Cao et al. (2019), we show how the EMD can perform ecologically-informed comparisons in the vegetation space. While more and more quantitative reconstructions of PFT distributions at regional scales have been published in recent years (e.g. the REVEALS-based studies by Githumbi et al. (2022) or Marquer et al. (2017)), we preferred using biomised data because biomes are currently the most-widespread format of publicly available continental-to-global scale syntheses of past vegetation changes (e.g. Binney et al. (2017) and Cao et al. (2019) for Eurasia during the last 40kyr, Prentice et al. (2000) for the Northern Hemisphere and Africa, and Marchant et al. (2009) in South America studying the mid-Holocene and Last Glacial Maximum, or Dowsett et al. (2016) for the mid-Pliocene). They also have a lower dimensionality than PFT data and provide, as such, a simpler context to explore the advantages of the EMD. Despite our focus on biomised data, it is important to stress that other categorical vegetation formats, such as pollen-based quantitative reconstructions as computed by REVEALS (Sugita, 2007) and Earth System Models, PFT affinity scores (*e.g.* Huntley et al., 2003; Allen et al., 2010; Henrot et al., 2017), or even the comparison of pollen percentages at the taxa level could have been used for our case studies. Finally, we discuss different research directions and fields where the EMD could be helpful.

## 2 The Earth Movers' Distance (EMD)

### 2.1 Concept and formalisation

The EMD is a distance metric measuring the minimal amount of work necessary to transform one entity into another. The general concept of the EMD algorithm can be most simply illustrated with the following everyday-life transportation problem: "What is the most cost-efficient way of transporting a fixed merchandise stock from $W$ warehouses to $R$ retailing shops?" (Levina and Bickel, 2001; Rubner et al., 2000). The problem can be reframed as: "How can the distribution of merchandise in the warehouses be transformed into the desired distribution of merchandise in the shops?". To solve this problem, we call $d_{i,j}$ the distance between warehouse $W_i$ and retailing shop $R_j$, $\omega_i$ the stock of merchandise at $W_i$ and $\rho_j$ the stock of merchandise needed at $R_j$. The EMD algorithm searches for the optimal combination of flows $f_{i,j}$ of merchandise (*i.e.* the amounts) to be moved between the warehouses and shops in a way that minimises the total cost C (*i.e.* the sum of how much is moved between locations multiplied by their distance).

$$C = \min_{i,j}\left(\sum_{i=1}^{W}\sum_{j=1}^{R} f_{i,j}.d_{i,j}\right)$$

**( 1 )**

with the constraints:

1. $f_{i,j} \geq 0, 1 \leq i \leq W, 1 \leq j \leq R$, *i.e.* the flow of merchandise between locations $W_i$ and $R_j$ is positive or null, which implies that the merchandise is moved from the warehouses to the shops, and not the opposite.

2. $\sum_{j=1}^{W} f_{i,j} \leq \omega_i$, *i.e.* the total amount of merchandise leaving warehouse $W_i$ to all the shops does not exceed its stock.

3. $\sum_{i=1}^{R} f_{i,j} \leq \rho_j$, *i.e.* the total amount of merchandise arriving at retailing shop $R_j$ from all the warehouses does not exceed its need.

4. $\sum_{i=1}^{R}\sum_{j=1}^{W} f_{i,j} = \sum_{i=1}^{R} \omega_i = \sum_{j=1}^{W} \rho_j$, *i.e.* the total amount of merchandise transported between warehouses and shops is equal to the initial amount of merchandise in the warehouses and the final amount of merchandise in the retail shops. The overall amount of merchandise is conserved.

Once the optimal flows are estimated, the EMD is calculated as follows (the minimal cost normalised by the sum of all flows):

$$EMD = \frac{\sum_{i=1}^{W}\sum_{j=1}^{R} f_{i,j}.d_{i,j}}{\sum_{i=1}^{W}\sum_{j=1}^{R} f_{i,j}}$$

**( 2 )**

Based on this formal definition, the transportation problem can be reframed in a broader context to become equivalent to finding the 'shortest' way of transforming one probability mass/density distribution into another (Levina and Bickel, 2001). With its flexibility, the EMD has been employed in a wide range of contexts, including, for instance, image retrieval algorithms (Rubner et al., 2000), the comparison of inorganic compositions (Hargreaves et al., 2020) or biomarker expression in cells (Orlova et al., 2016). To our knowledge, it has never been used to compare palaeoecological datasets.

## 2.2 The EMD applied to biomised data

### 2.2.1 Terminology

In this study, we propose to use the EMD to compare biome affinity score distributions from vegetation simulations and reconstructions. The transportation of merchandise becomes a transport of affinity scores between samples (*i.e.* a transformation of the vegetation composition of one sample into another). The concept of physical distance between entities (*e.g.* warehouses and shops) can be reframed as the ecological 'cost' of replacing a type of biome with another one. To ensure compatibility with constraint 4 of the previous section (the amount of 'merchandise' or 'affinity scores' is the same between the two entities compared), the affinity scores are normalised to sum to 1. This step is essential because most biomisation techniques do not ensure biome scores sum to a common target.

### 2.2.2 Definition of the weighting scheme (ecological distance)

We also use two different weighting schemes (*i.e.* the cost of replacing a biome with another one, the "$d_{ij}$" from Eq. 1 ) to illustrate how ecological knowledge can be introduced in such studies (Fig. 2). We use a "uniform" scheme where all the biome changes are given the same weight (EMD$_{uni}$) and an "ecologically informed" scheme (EMD$_w$), where differences are weighted based on vegetation structural differences (forest vs open landscape vs desert) and climate zone preferences (boreal vs temperate vs warm-temperate / subtropical). This dual definition of biome distance follows the work of Allen et al. (2020), in which each biome is assigned to one vegetation structure and one climate zone. In this study, we assume that the basal distance between two biomes with the same vegetation structure and climate zone is set to 0.5. Then, each difference in structure or climate zone adds an extra cost of 1 (*e.g.* moving affinity scores from a temperate forest to a warm desert costs 2.5; moving affinity scores from temperate forest to boreal forest costs 1.5). This simple weighting scheme illustrates how an ecologically-informed strategy can refine interpretations compared to the uniform scheme. Different research questions or settings could lead to using schemes with more detailed structural and climatic zone categories (*e.g.* Allen et al., 2020) or alternative weighting schemes based on, for instance, trait differences (*e.g.* Sato et al., 2021).

### 2.2.3 Rescaling the EMD to values between 0 and 1

The EMD calculated with normalised biome scores and a weighting scheme is a dissimilarity metric that varies between 0 (the two distributions are identical, and nothing needs to be moved) and the highest cost of that weighting scheme (here defined as $\max\limits_{i,j} d_{i,j}$ ). The highest distance can be reached when all the scores are transferred between the most different macro-environments and climate categories. In our ecologically-informed weighting scheme, this would correspond, for instance, to the transformation of a pure boreal forest composition into a warm temperate desert, or *vice-versa*, and would be given a weight of 4.5 (Fig. 4). The EMD can be normalised by the highest possible cost for a given weighting scheme:

$$EMD_n = \frac{EMD}{\max\limits_{i,j} d_{i,j}}$$

Unlike other metrics for which expert-elicited quality thresholds have been proposed (*e.g.* kappa statistics, Altman (1990) or Landis and Koch (1977)), no expert-based quality assessment exists for the $EMD_n$. In fact, defining such a quality scale could be counter-productive, as many study-dependent factors influence the range of values the $EMD_n$ will take in each study. These include: 1) the number of biomes to compare (more biomes usually lead to higher distances), 2) the definition of the weighting schemes (the $EMD_n$ is inversely proportional to the highest cost), or 3) the data structure of the entities being compared (compare the $EMD_n$ ranges in the data-data (same structure) and data-model (unary data compared to multidimensional data) comparison applications below for concrete examples). Comparing EMD values between studies should, therefore, always be done carefully.

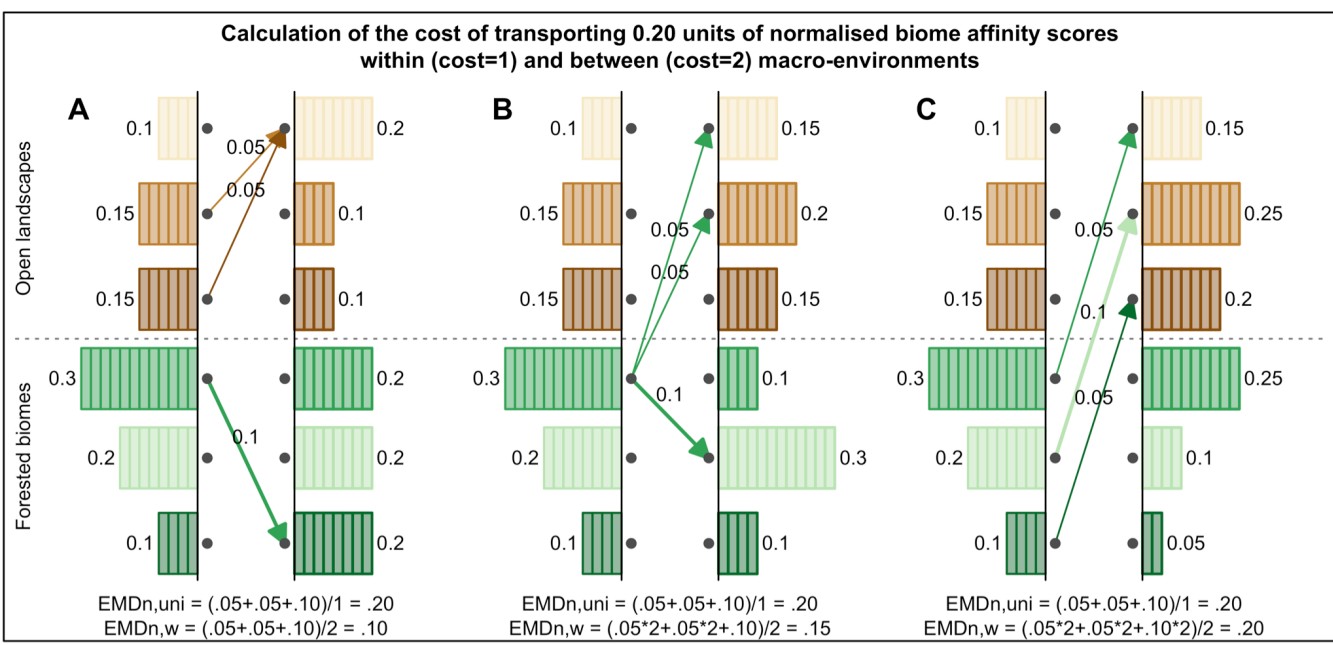

**Figure 2. Calculation of the $EMD_n$ for three different scenarios in which 0.20 units of normalised biome affinity scores are transported: (A) all changes are within the same macro-environments, (B) changes are both within and between macro-environments, and (C) all changes are between macro-environments. The use of ecologically-informed costs (here only two costs: 1 or 2) leads to three distinct values for $EMD_{n,w}$ (0.1, 0.15, 0.2), while the uniform approach considers that all three changes are equivalent ($EMD_{n,uni}$ = 0.2).**

### 2.3 Implementation of the EMD in the 'paleotools' R package

To facilitate access to the EMD, we developed an R package called paleotools using R (R Core Team, 2022) and the *devtools* package (Wickham et al., 2020). To calculate the EMD, *paleotools* includes a wrapper function of the *emd()* function from the *emdist* R package (Urbanek and Rubner, 2022). In addition, we also developed two statistical tests, *signif_threshold()* and *signif_struct()*, to overcome the absence of 'quality' thresholds. The package is accessible from https://github.com/mchevalier2/paleotools.

**Test 1: Considering the parameters of a study, can two samples be considered similar?** This test is inspired by the Monte Carlo simulation designed by Sawada et al. (2004) to identify analogue samples from a large and heterogeneous collection of modern pollen samples. The underlying idea is to determine a distance threshold that is unlikely to have occurred by chance (Simpson, 2007). To do so, a large number of pairs of biome score distributions are randomly drawn from a data collection, and their EMD is calculated. This results in a distribution of EMD values derived from the randomised comparison of biome score distributions. The EMD value corresponding to a certain percentile of that distribution (*e.g.* the 5[th] percentile) can be used as an empirical estimate of a similarity threshold. EMD values below/above that threshold correspond to comparisons of similar/different samples. Importantly, this test cannot determine if two samples represent the same biome. Because biomes are, by definition, broad vegetation units, two samples can be statistically different and be characteristic of the same biome. In such cases, the statistical difference suggests they likely occupy a different position in that biome's vegetation and/or climate spaces. For instance, vegetation samples taken from the cold and warm ends of the temperature range experienced by the temperate forest biome are likely to be statistically different while still being representative of temperate forests. This test can only be used if hundreds of affinity score distributions representative of various environments are available to estimate the randomisation distribution. In addition, this type of threshold is only valid for a given study area and/or research question. The test is called *signif_threshold()* in *paleotools,* and its use and interpretation are illustrated in Section 4.

**Test 2: Considering the parameters of a study, are the data and the simulation (or modern observations) displaying similar spatial patterns?** This second test aims to determine if the mean EMD value obtained when comparing a simulated (or observed) vegetation map with a large collection of pollen-based affinity score distributions is smaller than expected when comparing two datasets with different spatial patterns. This test is performed in two steps. First, the data are shuffled (each affinity score distribution is randomly assigned to one of the modelled values corresponding to a sample location), and the resulting mean EMD across all locations (*i.e.* spatial mean) is calculated. This is repeated several times to estimate the distribution of spatial mean EMD values under the assumption that the spatial structure in the data differs from the spatial structure of the simulation (null hypothesis). The 5[th] percentile of that distribution (any other significance threshold could be used depending on the research question) represents the threshold to reject the null hypothesis (alternative hypothesis: the data and the simulation have similar spatial structures). Then, the uncertainty of the observed EMD value is estimated by measuring the intra-sample variability. To do so, a second EMD distribution is estimated by bootstrapping, *i.e.* randomly sampling the same number of biome samples with replacement (some samples are selected many times and others excluded) and calculating the EMD of this bootstrapped dataset with the observed/simulated vegetation map. To determine if the data and the simulation display the same spatial pattern, the 95[th] percentile of the bootstrapped distribution is compared with the 5[th] percentile of the distribution of the null hypothesis (one-sided test). If the former is larger than the latter, the null hypothesis is rejected, and the spatial structure of the simulated and reconstructed biomes is considered similar. Efron and Tibshirani (1994) recommend

performing at least 200 repetitions to estimate the bootstrapped and null hypothesis distributions. This test is called *signif_struct()* in *paleotools,* and its use and interpretation are illustrated in Section 5.

## 3 Data

### 3.1 Pollen and biome reconstructions

40    To illustrate the use and strength of the EMD for palaeoecological studies, we use the pollen-based biome reconstructions presented by Cao et al. (2019). The dataset covers the entire Northern Hemisphere extratropics. Here, we restrict it to the Euro-Mediterranean Basin, where the quality and quantity of pollen records are ideal for testing the EMD in various conditions (Fig. 3). The pollen data were extracted from the European Pollen Database in June 2017, and a total of 1347 records fall within our study area. The biomisation strategy employed by Cao et al. (2019) follows the biomisation tables presented by Binney et al.

45    (2009) and Bigelow et al. (2003), and the algorithm of Prentice et al. (1996). 13 distinct biomes can be theoretically reconstructed across the study area (Table 1).

| Macro-environments | Climatic zone | Mega-biome (Dallmeyer et al., 2017) | Biomes from BIOME4 | Euro-Mediterranean biomes from pollen (Cao et al., 2019) |
|---|---|---|---|---|
| Forests | Temperate | Temperate forest / Woodland (TEDE) | Temperate deciduous forest | Temperate deciduous forest |
| | | | Temperate conifer forest Temperate sclerophyll woodland | |
| | Warm temperate / Subtropical | Warm forest (WARF) | Warm mixed forest | |
| | Boreal | Cold/Cool forest (COCO) | Cool mixed forest | Cool evergreen needle-leaved forest |
| | | | Cool conifer forest | Cool-temperate evergreen needle-leaved forest |
| | | | Cold mixed forests | Cool mixed forest |
| | Boreal | Evergreen Taiga (TAIG) | Evergreen taiga / montane forest | Cold evergreen needle-leaved forest |
| | Boreal | Deciduous Taiga / Boreal woodland (BORW) | Deciduous taiga / Montane forest | Cold deciduous forest |
| | | | Open conifer woodland | |
| | | | Boreal parkland | |
| Herbaceous / Open landscapes | Temperate | Shrubland (SHRU) | Temperate xerophytic shrubland | Temperate xerophytic shrubland |
| | | | Tropical xerophytic shrubland | |
| | Temperate | Grassland (GRAS) | Tropical grassland | Temperate grassland |
| | | | Temperate grassland | |
| | Boreal | Tundra (TUND) | Steppe tundra | Graminoid and forb tundra |
| | | | Shrub tundra | Low and high shrub |
| | | | Dwarf shrub tundra | Erect dwarf-shrub tundra |
| | | | Prostrate shrub tundra | Prostrate dwarf-shrub tundra |
| | | | Cushion forb lichen moss tundra | Cushion-forb tundra |
| Deserts | Warm temperate / Subtropical | Desert (DESE) | Desert | Desert |

**Table 1. Biome to mega-biome to macro-environments assignments following Dallmeyer et al. (2017) for the simulated biomes and Cao et al. (2019) for the reconstructed biomes.**

## 3.2 Climate and vegetation simulations

We use the vegetation simulations presented by Cao et al. (2019). These simulations were derived from the biome model BIOME4 (Kaplan et al., 2003) in the version adapted by Dallmeyer et al. (2017). BIOME4 calculates the equilibrium biome distribution for 28 potential biomes using a prescribed climate and taking biogeographical and biogeochemical processes into account (Kaplan, 2001; Kaplan et al., 2003). Of these 28 biomes, 21 were observed in our study area for at least one time interval of the available simulations (Table 1). Input variables are climatological monthly mean temperature, cloud cover and precipitation, the climatological mean absolute minimum temperature of the year, atmospheric $CO_2$ concentration, and physical properties of the soil such as water-holding capacity and percolation rates. The results are provided as one single biome per grid cell, hereafter called 'unary biome estimate'.

In the simulations used here, BIOME4 has been forced by climate simulations conducted in the coupled general circulation model Community Earth System Models (COSMOS) in the spatial resolution T31 (~4°x4° on a gaussian grid). COSMOS was developed at the Max Planck Institute for Meteorology. It consists of the general circulation model for the atmosphere ECHAM5 (Roeckner et al., 2003) coupled with the land-surface model JSBACH (Brovkin et al., 2009) and the ocean model MPIOM (Marsland et al., 2003). An anomaly approach has been used to prepare the climate input data for the biome model and reduce systematic model biases, for instance, due to the coarse spatial resolution of the model in which the orography is strongly smoothed. For this purpose, the difference between the climate simulated for a particular time slice and the pre-industrial reference climate has been calculated, bilinearly interpolated to a regular 0.5°x0.5°grid and added to observations (here: CRU-TS3.10 data, Harris et al., 2014). Five timeslices are available, i.e. 21ka and 14ka (Zhang et al., 2013), 9ka and 6ka (Wei and Lohmann, 2012), and 0ka (Wei et al., 2012). Further details and global boundary conditions of the climate simulations are described in Dallmeyer et al. (2017) and Tian et al. (2018).

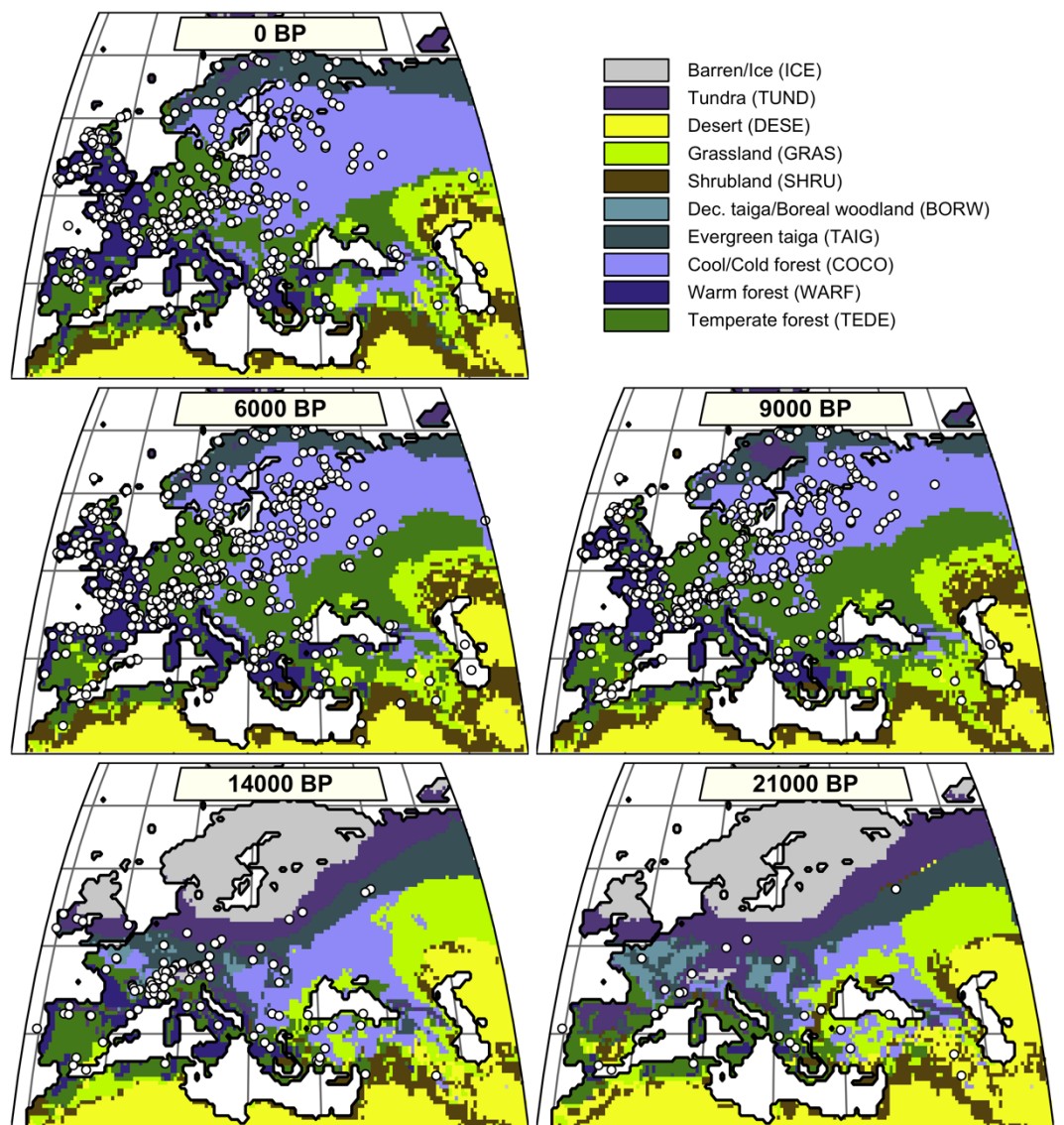

**Figure 3. Distribution of the simulated mega-biomes at 0, 6, 9, 14 and 21 ka. Each pollen-based biome estimate is represented as a white dot and corresponds to a distribution of biome affinity scores, as illustrated in Fig. 1.**

### 3.3 Harmonisation of the biome reconstructions and simulations

Since the definition of biomes was slightly different in the two datasets, the biome reconstructions and simulations were harmonised with the 'mega-biome' scheme of Dallmeyer et al. (2017) to enable direct comparison. This scheme is a global classification tool composed of 12 levels, which allows the grouping of biomes into higher-order vegetation classes. Nine mega-biomes were observed across the study area (Fig. 3 and Table 1). The harmonisation of the model results at the mega-biome level was straightforward because they were only available as unary biome estimates for each grid cell. Each grid cell

was assigned to the mega-biome corresponding to its biome (Table 1). Harmonising the pollen data was more challenging because the data were only available as arrays of biome scores. Since many taxa are part of multiple biomes, adding the scores of the different biomes belonging to the same mega-biomes would lead to overestimating the mega-biome scores (*i.e.* the weight of some taxa would be accounted for several times). Re-running the biomisation algorithm would have thus been necessary to obtain exact mega-biome scores (replacing the 'plant functional type to biome' table with a 'plant functional type to mega-biome' table in the biomisation algorithm). However, not all the required data were available. For simplicity, we assumed the mega-biome scores could be defined by the highest score of all their composing biomes (see Table 1 for the detailed biome composition of each mega-biome). This solution is imperfect and underestimates the actual scores. Still, we believe this simplification is sufficient for the purpose of this study, which is to illustrate how the EMD can be used in data-data and data-model comparison studies and not generate/evaluate new data. Finally, the mega-biomes were grouped into three macro-environments ('forested environments', 'herbaceous/open landscapes', or 'deserts') and three climatic zones ('boreal', 'temperate', or 'Warm temperate/subtropical') to define the weights used to calculate the $EMD_{n,w}$ (Table 1; Fig. 4).

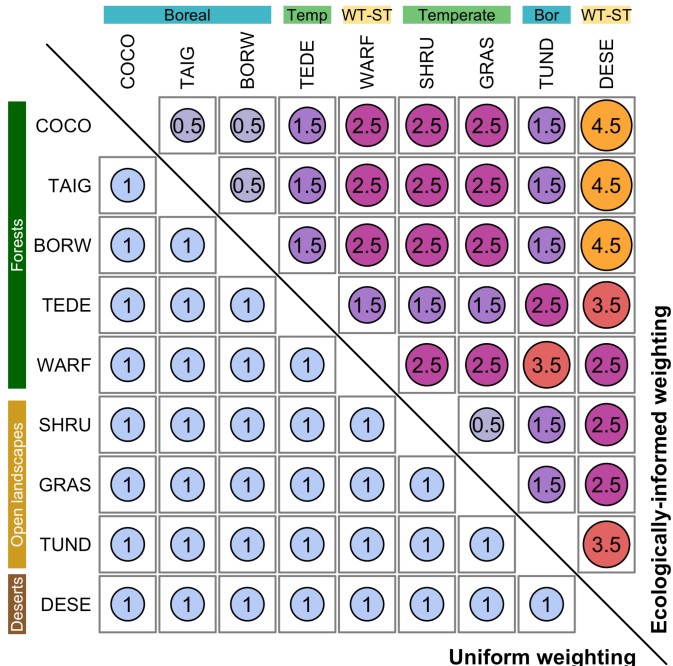

**Figure 4. The two penalty matrices used in this study. The lower (upper) triangle in blue (purple to yellow colour gradient) represents the uniform (ecologically-informed) weighting, respectively. In both cases, the diagonal of the matrix contains 0s. The biome acronyms are defined in Table 1.**

# 4 Data-data comparison: EMD vs mega-biomes with the highest affinity score

## 4.1 Discrimination between mega-biomes

We perform a series of data-data comparison case studies to evaluate the performance of the $EMD_{n,w}$ compared to analyses based on reconstructions of mega-biomes with the highest affinity score only. First, we analyse the $EMD_{n,w}$ values calculated between 2000 randomly selected mega-biome samples, irrespective of their ages (past and modern samples were pooled together). The resulting ~2 million unique pairwise comparisons are grouped according to the agreeing or disagreeing status of their mega-biome with the highest affinity score. If two samples have the same mega-biome with the highest affinity score X, as in Fig. 1A, the pair is labelled as 'Mega-biome X'. If they differ (Mega-biome X and Mega-biome Y, as in Fig. 1B), the pair is labelled as 'Mega-biome X with other biomes' and 'Mega-biome Y with other biomes'. Note that this labelling does not imply that the vegetation necessarily belongs to the mega-biome X but only that mega-biome X has the highest affinity score. The two resulting $EMD_{n,w}$ collections for Mega-biome X (*i.e.* 'Mega-biome X' and 'Mega-biome X with other mega-biomes') are interpreted as the intra-mega-biome and inter-mega-biome $EMD_{n,w}$ variability distribution of samples with Mega-biome X as the mega-biome with the highest affinity score. The results for the five most abundant mega-biomes across the study area and the $EMD_{n,w}$ are summarised in Fig. 5 (see Appendix 1 for the same analysis with the $EMD_{n,uni}$).

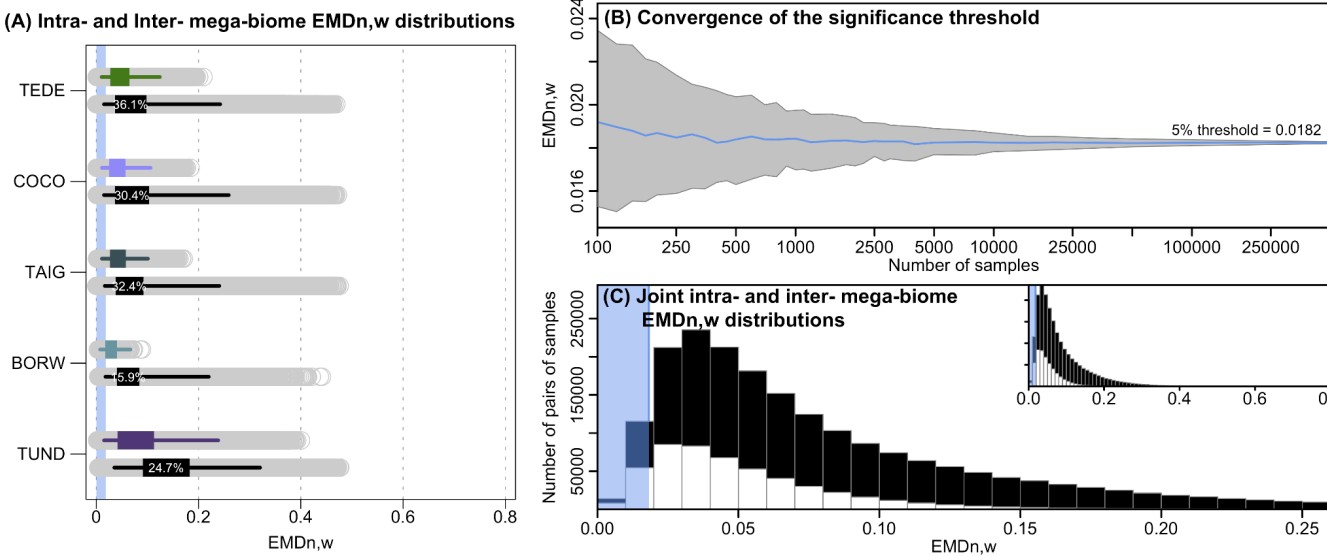

**Figure 5.** (A) Distribution of intra- (coloured) and inter- (black) mega-biome $EMD_{n,w}$ distributions. For all mega-biomes, the top/coloured boxplot represents the distribution of the pairwise distances of all the samples with the same mega-biome with the highest affinity score, and the bottom/black boxplot represents the $EMD_{n,w}$ distributions of these samples with different mega-biomes with the highest affinity score. The box of the boxplot represents the 25-75% interval (interquartile range), and the whiskers represent the 2.5-97.5% interval. The percentages indicate the proportion of samples where the $EMD_{n,w}$ of the inter-mega-biome distribution is lower than the intra-mega-biome distribution (estimated from 10,000 bootstrapped pairs of samples drawn from the intra- and inter-mega-biome $EMD_{n,w}$ distributions). The higher the percentage, the higher the overlap between the two distributions. (B) Evolution of the estimation of the $EMD_{n,w}$ threshold as a function of the number of samples selected. Note the log scale of the x-axis. (C) Distribution on the intra- (white) and inter- (black) mega-biome $EMD_{n,w}$ (all mega-biomes pooled together). The blue band

 **in (A) and (C) represents the range of $EMD_{n,w}$ values that characterise statistically similar samples based on our first statistical test (estimated in (B)).**

These data are also used to explore how the proposed statistical similarity test performs (Test 1, Section 2.3). We test the 'stability' of the significance threshold as a function of the number of $EMD_{n,w}$ values available. At the scale of Europe, the $EMD_{n,w}$ threshold for a significant similarity at 5% is ~0.0182 (Fig. 5B). While this value can be correctly estimated on average from a few samples, its variability is, however, high when only a limited number of samples are selected. Undersampling the data (or small-sized datasets) can thus lead to an increased risk of mistakenly rejecting or accepting the null hypothesis (H0: The two samples are dissimilar). Here, our results suggest that considering about 10,000 $EMD_{n,w}$ values, which corresponds to all the pairwise comparisons between about 140-150 independent samples, is necessary to obtain stable thresholds. The results of this similarity test are always relative to the size of the study area, wherein small-scale studies will have smaller EMD thresholds because the samples will be more similar on average. Each threshold is thus study-specific and should not be employed in a different context.

For the five biomes selected here, the mean $EMD_{n,w}$ of the intra-mega-biome distributions are smaller than the mean $EMD_{n,w}$ of the corresponding inter-mega-biome distributions and large intra-mega-biome $EMD_{n,w}$ values are not observed for most mega-biomes, except for tundra (TUND). This result is coherent and expected, as the mega-biome with the highest affinity score estimate is a summary measure that extracts the dominant signal of the data. However, comparisons of the intra-mega-biome $EMD_{n,w}$ distribution with the inter-mega-biome $EMD_{n,w}$ distribution highlight a substantial overlap. Many pairs of samples from the inter-biome distributions have very low $EMD_{n,w}$, suggesting strong similarities in their relative mega-biome compositions despite having different mega-biomes with the highest affinity scores (comparisons similar to the example in Fig. 1B). In the 'extreme' case of temperate deciduous forests (TEDE), about one-third of the pairs from TEDE's inter-mega-biome $EMD_{n,w}$ distribution has a smaller $EMD_{n,w}$ than pairs from TEDE's intra-mega-biome $EMD_{n,w}$ distribution (estimated from a random drawing from each group; Fig. 5). This considerable overlap between the inter- and intra-mega-biome distributions can be further illustrated with the statistical test we designed to determine if two samples can be considered similar (Test 1). Of all the significant pairwise comparisons (all mega-biomes included), only one-half corresponds to comparisons of samples with identical mega-biome with the highest affinity score estimates (Fig. 5C). Therefore, these results demonstrate that while the highest affinity score approach produces good results on average, fine-scale details of the vegetation structure are missed in some comparisons when samples are solely labelled by the mega-biome with the highest affinity score.

## 4.2 Characterising mega-biome changes in space and time

With the second data-data comparison study, we show how the more gradual response of the $EMD_{n,w}$ to changes in mega-biome affinity score distributions can refine vegetation change interpretations through time and space (Fig. 6). When mega-biome reconstructions are represented by the mega-biome with the highest affinity score only, oscillations between different mega-biome unary estimates can happen, as a result of minor changes in the affinity scores that cause an apparent oscillation

between unary estimates when the multidimensional data are reduced to univariate estimates (as could happen between the two samples on Fig. 1B). This is further illustrated by the mega-biome reconstruction of Cao et al. (2019) from the pollen record Lago Piccolo di Avigliana (Finsinger et al., 2011; Finsinger and Tinner, 2006; Fig. 6A-C). For this record, we calculate 1) the $EMD_{n,w}$ of all the samples with the top sample to measure the broad trends of mega-biome divergence over time relative to modern-day and 2) the sample-to-sample $EMD_{n,w}$ to measure the variability of vegetation between temporally contiguous samples. We also used the similarity significance threshold (EMD = 0.0182) defined in the previous section since the settings of the two analyses are the same.

Significant vegetation changes are evident in the record, with all the samples older than 1000 BP being dissimilar to the top sample. Representing the data by the mega-biomes with the highest affinity score suggests high vegetation instability over time (52 changes for 321 samples). However, if these mega-biome shifts are analysed with the EMD, most sample-to-sample changes are associated with statistically similar samples. In particular, the mean differences between contiguous samples that trigger a change in the mega-biome with the highest affinity score ($\overline{EMD_{n,w}}$ = 0.026, sigma = 0.016, n = 51) are not statistically different from the mean changes between samples that do not ($\overline{EMD_{n,w}}$ = 0.024, sigma = 0.015, n = 269; t-test p-value = 0.39). As opposed to the representation based on mega-biomes with the highest affinity score that suggests a similar sample-to-sample vegetation variability across the record, the sample-to-sample $EMD_{n,w}$ values (Fig. 6B) suggest that vegetation changes were relatively slower before ~7,000 BP ($\overline{EMD_{n,w}}$ = 0.020, sigma = 0.010, n = 121) and since ~1500 BP ($\overline{EMD_{n,w}}$ = 0.024, sigma = 0.018, n = 16) and more intense in between ($\overline{EMD_{n,w}}$ = 0.027, sigma = 0.017, n = 184). This example illustrates how the type of representation chosen for the data can influence interpretations. In this case, the oscillations visible in the unary biome estimates are mainly a visual artefact resulting from simplifying the data to single estimates instead of looking at the entire distribution of mega-biome scores. In contrast, the statistical test provides a more robust way to select time steps with significant biome composition changes.

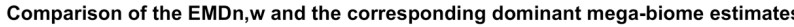

**Figure 6. Comparison of the EMD$_{n,w}$ and the corresponding mega-biomes with the highest affinity scores (A–C) in time and (D–E) in space. (A) 'mega-biome with the highest affinity score' reconstruction for a pollen record from northern Italy (Cao et al., 2019; Finsinger et al., 2011; Finsinger and Tinner, 2006). (B) EMD$_{n,w}$ calculated between contiguous pairs of samples, highlighting that vegetation changes that trigger a change in the mega-biome with the highest affinity score are not different from the changes that do not. (C) EMD$_{n,w}$ of the biome scores compared to the top sample, highlighting significant vegetation changes across time. The significance threshold at 5% (blue band) was derived from the random sampling of 2000 pairs of Holocene samples across Europe. (D–E) Mapping of the EMD$_{n,w}$ of all the regional samples compared to the mega-biome reconstruction at the location indicated with a red diamond at 0 BP (D) and 6000 BP (E).**

Similar smooth transitions can be observed for the variability across space, where the spatial granularity of the data is much lower than what is suggested by considering only mega-biomes with the highest affinity scores (Fig. 6D and E). Many neighbouring samples characterised by distinct mega biomes with the highest affinity scores are, in fact, similar according to the EMD$_{n,w}$ (*e.g.* the small dots of different colours near the target sample in Fig. 6D and E). In general, the size of the dots (*i.e.* the EMD$_{n,w}$) increases with distance to the target sample or with a higher elevation, such as in the Alpine and Carpathian regions. The mean EMD$_{n,w}$ values at 6,000 BP ($\overline{EMD_{n,w}} = 0.056$, sigma $= 0.042$, n $= 307$) is much lower than the mean EMD$_{n,w}$

modern values ($\overline{EMD_{n,w}}$ = 0.104, sigma = 0.068, n = 268), and their distribution in space is much more regular. Determining why these differences exist is beyond the scope of this paper. However, it could be related to the influence of humans on modern environments (*e.g.* deforestation and opening of the landscapes) or different climate conditions.

## 5 Data-Model comparison: Evaluation of vegetation simulations

Data-model biome comparisons are commonly based on comparisons of unary biome estimates from models, pollen assemblages or modern observations. In such cases, the number of agreeing and disagreeing pairings is used to measure the accuracy, and the results are reported in contingency tables. These tables are ultimately analysed using different indices (*e.g.* the Kappa statistics, Cohen (1960)). As shown in the previous section, this type of data simplification is suboptimal because the information on the distribution of biome affinity scores cannot be accounted for. For example, if the matching biome is the biome with the second-highest affinity score, the strength of the mismatch should not be the same as if it were the biome with the lowest affinity score. To illustrate the advantage of the EMD coupled with ecologically-informed weights, we reproduced the data-model comparison of Cao et al. (2019), who used the Kappa statistic (among other metrics) to evaluate the similarity of the patterns displayed by the data and models. The main results are summarised in Table 2 and represented in Fig. 7.

| Time interval | Accuracy (correct/total) | Kappa statistic | $EMD_{n,uni}$ | $EMD_{n,w}$ |
|---|---|---|---|---|
| 0 BP | 77 / 368 | 0.04 | 0.853 | 0.254 (NS) |
| 6000 BP | 106 / 422 | 0.06 | 0.817 | 0.227 (NS) |
| 9000 BP | 85 / 292 | 0.08 | 0.790 | 0.199 |
| 14000 BP | 21 / 100 | 0.00 | 0.929 (NS) | 0.647 (NS) |
| 21000 BP | 4 / 28 | 0.00 | 0.908 (NS) | 0.593 (NS) |

**Table 2: Summary statistics of the data-model comparison. The accuracy and Kappa statistics are reported by Cao et al. (2019), and the EMD values result from our statistical tests. Both are reported as the mean of all values across the study area. The Kappa statistics and the EMD are similarity and dissimilarity measures, respectively. A value of 1 (0) is the best (worst) score for the Kappa statistics and the worst (best) score for the EMD. Non-significant tests are labelled with (NS).**

One limitation of this study is that we compare pollen-derived multidimensional biome score distributions with model-derived unary biome distributions, where all the mass is concentrated on one single biome. This fundamental difference in the data structure of the two entities being compared means that reaching an EMD of 0 is highly unlikely because obtaining such a concentrated distribution of biome scores from pollen data is nearly impossible. In our example dataset, the highest biome affinity score for a sample is generally in the range of 0.20-0.35 (Fig. 1). These values imply that even if the biome with the highest affinity score of a pollen sample matches the simulated biome of the corresponding grid cell, the $EMD_{n,uni}$ will have, in general, a value of about 0.65-0.80 because all the other biome affinity scores will have to be "moved" to the biome category with the highest affinity score. The same principle applies to the $EMD_{n,w}$, but calculations of a 'best case scenario' range are less direct due to the penalty matrix. This explains why the absolute EMD values of this data-model comparison are much

higher than the EMD values calculated in the previous data-data comparison based on the comparisons of datasets with similar distribution structures. Nevertheless, this technical limitation does not impede using the EMD for data-model comparisons.

Among all timeslices, models and data are most consistent at 9 ka according to the three evaluation indices (Table 2). The overall ranking of the five data-model comparisons based on the $EMD_{n,w}$ and $EMD_{n,uni}$ is also consistent with the Kappa
statistics and accuracy of Cao et al. (2019) (Table 2). We used the statistical test defined in Section 2.3 (Test 2) with both the $EMD_{n,w}$ and $EMD_{n,uni}$ to determine if the spatial patterns of the simulated and reconstructed biomes for the five timeslices (0, 6, 9, 14 and 21 ka) were similar. The results indicate that the data-model comparisons for the 9 ka timeslice are significant in both cases, while those for timeslices at 14 ka and 21 ka are not. Interestingly, the comparisons at 0 ka and 6 ka are significant with the $EMD_{n,uni}$ and not significant with the $EMD_{n,w}$. These contrasting results can be explained by two types of differences.
First, macro-environmental differences are observed, mainly at 0 ka, when many mismatches correspond to pollen samples with tundra as their biome with the highest affinity score (TUND, Fig. 7) and when the model instead simulates cold forest environments (either TEDE or COCO). The mismatches at 0 ka and 6 ka are also caused by climatic differences between the type of forests simulated (warm forests) and reconstructed (temperate forests) in western Europe. By definition of the penalty matrix and the 'ecologically-informed' ranking of mismatches (Fig. 4), replacing forests with more open landscapes or changes
in the climate types are more penalised in $EMD_{n,w}$. This difference tips the test result from significant without the weights (the two datasets have a similar spatial structure if all biome differences are considered equal) to non-significant when the weights are included (their spatial structure is different if we assume that replacing a forest with more open landscapes or a temperate forest with a warm forest is a large ecological change).

While identifying the reasons underlying these mismatches is beyond the scope of this paper, we can hypothesise that the difference in landscape openness at 0 ka could be related to human land use. In addition, the assignment of biomes in the biome model is primarily controlled by climatic conditions, while other environmental conditions, such as soil conditions, also influence natural vegetation. For instance, wetlands or peatlands may result in open landscapes, even if the climate conditions could support forests. In contrast, the spatially and temporally consistent forest mismatch in Western Europe during the
Holocene points towards a different definition of warm and temperate forests in the simulated and reconstructed data. The model explicitly excludes the temperate broadleaved tree PFT in the temperate forest biome, while it is included in the temperate and warm temperate forest biome in the reconstructions.

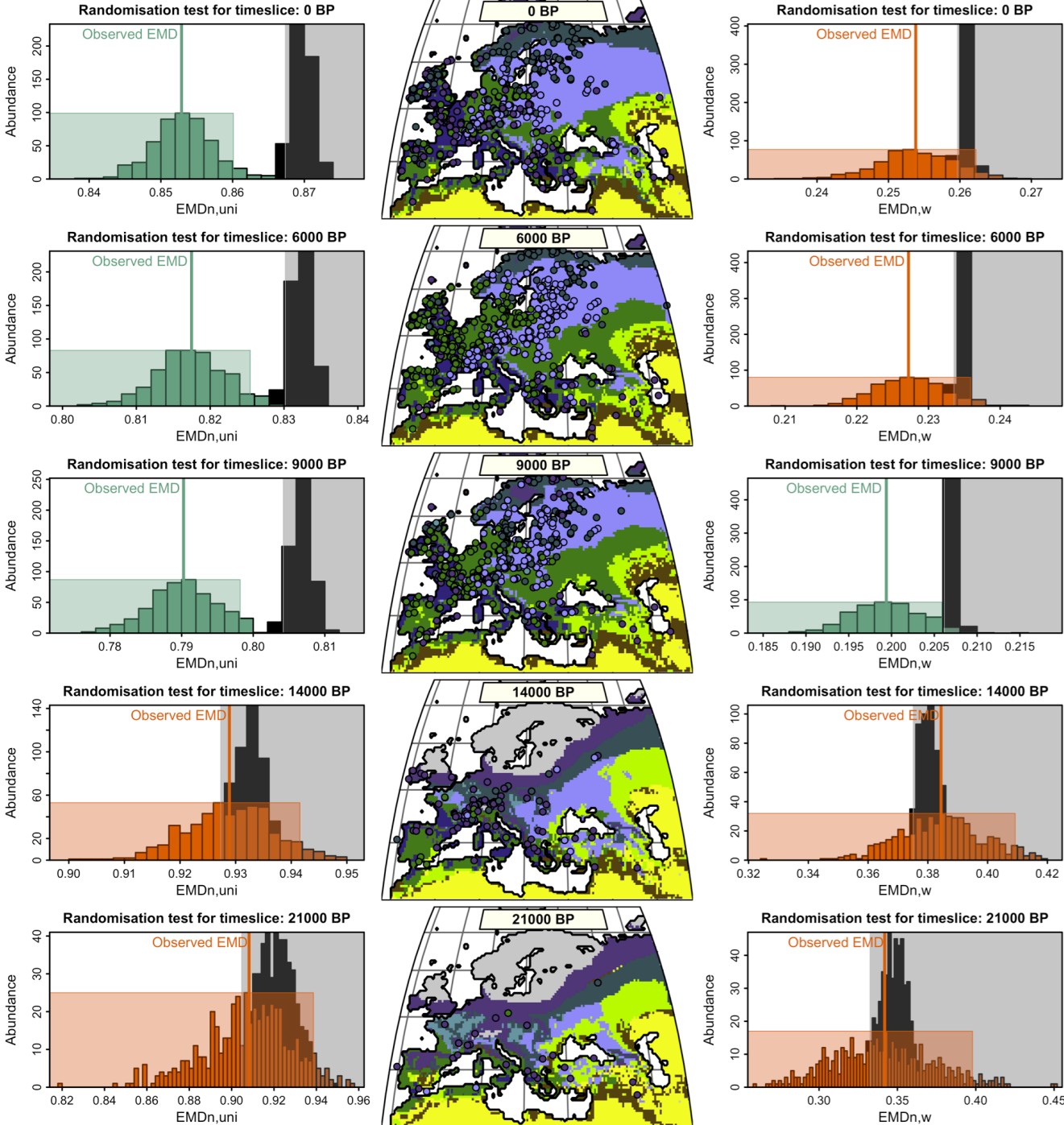

**Figure 7. Data-model comparisons for the five simulated timeslices using either $EMD_{w,uni}$ (left) or $EMD_{n,w}$ (right). (centre) The mega-biomes with the highest affinity scores derived from the pollen data are plotted over the simulated unary mega-biome estimates. (left/right) Statistical test evaluating the degree of similarity between the reconstructed and simulated mega-biomes. The black histogram represents the distribution of EMDs under the null hypothesis (the spatial distributions of the two datasets are different).**

**The coloured histogram represents the uncertainty distribution of the observed EMD. The null hypothesis is rejected when the black and coloured rectangles do not overlap (the rectangles are defined based on a 5% significance threshold and 500 repetitions). Green/Orange means the null hypothesis is rejected/accepted, and the two datasets have a similar/different spatial structure.**

These results also demonstrate that propagating significance thresholds across studies should be avoided since significance levels are directly determined by the data and the parameters of the study. For example, the $EMD_{n,uni}$ value of 0.853 is significant for the data-model comparison at 0 ka with the uniform penalty matrix, while the $EMD_{n,w}$ value of 0.254 is not when using our ecologically-informed matrix for the same time interval. As explained earlier, this behaviour is expected and precludes defining a global significance threshold for the EMD.

## 6 Perspectives

The case studies presented in the previous sections illustrate how using a continuous metric, as opposed to a binary assessment of similarity, can help refine interpretations of data-data comparisons and facilitate a better understanding of vegetation dynamics through time (Fig. 6A-C) and space (Fig. 6D-E). The EMD also proved to be a powerful tool for performing statistically robust data-model comparisons, despite using unary distributions for the simulated data (Fig. 7). Our case studies demonstrated that 1) while interpretations only based on mega-biomes with the highest affinity scores tend to be correct on average, they miss fine-scale details of the data (Fig. 5), and 2) the simplification to unary estimates can add noise to vegetation reconstructions (*e.g.* temporal oscillation of the mega-biomes with the highest affinity scores, Fig. 6) that may be difficult to interpret because the underlying data changes smoothly. However, these examples represent only a fraction of the applications where the EMD could be helpful. For example, the bootstrapping approach used to estimate the uncertainties of the observed EMD value could be used to compare if the data-model agreement of one time slice is more robust than another (*e.g.,* is the data-model agreement at 9K statistically better than the data-model agreement at 6K?), or similarly, if the data agree more with the simulation of one specific model rather than another one.

The EMD could also be used to optimise the biomisation schemes themselves. Creating such schemes often requires tuning multiple parameters in parallel while evaluating the results with modern vegetation maps. Due to the sensitive nature of unary mega-biome estimates, small parameterisation changes can easily change one mega-biome with the highest affinity score into another (Fig. 1B), which can strongly impact Kappa statistics and other binary indices. Using the EMD would allow for a smoother evaluation of the impact of changing some parameters. The penalty matrix could also become a parameterisable variable for biomisation studies. The one used in this study is based on simple ecological considerations based on structural and climatic zone changes, but more complex, data-informed distance matrices could be designed by, for instance, calculating (some form of) inter-mega-biome distance in the climate and/or vegetation spaces, integrating plant traits ecological distance (e.g. Sato et al., 2022) or modelling the probability of mistaking one biome for another using independent calibration data.

Developing such alternative matrices is, however, complex as their stability in time and space should also be assessed before being used.

95 Despite its simplicity, our categorical penalty matrix already adds a level of refinement that is absent from most other biome comparison techniques. Similar "temporally-stable" matrices could also be defined to study pollen data at different taxonomical resolutions. For example, recent and ongoing work on vegetation cover reconstructions with the REVEALS model or the use of PFT affinity scores provides new avenues comparing the resulting PFT distributions with simulated PFT distributions based on coverage fraction, net primary production or leaf area index (e.g. Huntley et al., 2003; Allen et al., 2010;

00 Marquer et al., 2017; Henrot et al., 2017). Penalty matrices on the level of taxa resolved in pollen records could also be developed to integrate the EMD into pollen-based climate reconstruction algorithms since many of the existing techniques, such as the Modern Analogue Technique (Overpeck et al., 1985), are based on the direct comparison of pollen samples (Chevalier et al., 2020). In addition to measuring their statistical differences as is currently done, an EMD-based definition of pollen analogues would also include the ecology of the taxa so that a well-designed penalty matrix could refine the climate

05 reconstructions.

As with most distance metrics, the EMD only measures how dissimilar two samples are and does not provide direct information on the type of (multidimensional) direction of differences. For example, the EMD cannot tell whether sample A is more forested than sample B. It can only quantify how different samples A and B are. This is similar to the binary evaluations of

10 biomes with the highest affinity score. However, while it is common practice to characterise the direction of change by analysing the properties of the compared datasets separately, the computation of the EMD could offer more direct insights through the "optimal flows" that "transport" the affinity score distribution of sample A to the affinity score distribution of sample B (see Sect. 2.1). The optimal flows, which minimise the transport cost, could be written as a transport matrix. Therefore, this transport matrix would contain information on the (multidimensional) direction of the mismatch between

15 samples. However, quantifying and interpreting these flows is challenging because (a) the optimal flows are not necessarily unique (in fact, they will rarely be unique in the case of uniform weights since, in this case, all flows have the same cost), and (b) the form of the transport matrix depends on the penalty matrix and thus the level of ecological complexity implemented in the penalty matrix. As such, the ecological interpretability of transport matrices could be another advantage of the EMD compared to other metrics. Therefore, we believe that methods to interpret the "optimal flows" should be explored in future

research.

Finally, it is essential to emphasise that the EMD, as presented in this study, is not limited to vegetation studies. It can be used with any form of discrete palaeodata (*i.e.* ordinal and categorical) from different disciplines, including but without being limited to, all palaeoecological datasets (*e.g.* chironomids, foraminifera, rodents, *etc.*), geochemical datasets (*e.g.* n-alkane

distribution from terrestrial or marine sediments), or archaeological datasets (*e.g.* lithics and tools from archaeological

deposits). More generally, while raw data counts with a different total number of fossils/artefacts cannot be directly compared with the EMD, their percentages always can because they sum to 100. Said differently, any two samples can be compared with the EMD, provided they have the same total mass.

## 7 Conclusion

Comparisons of discrete palaeoclimatic vegetation data are often based on the co-evaluation of their best estimates. While based on sound principles, this approach has limitations, particularly regarding the impossibility of accounting for the multidimensionality of the data. This paper proposes to replace the binary metrics commonly used to perform data-data or model-data vegetation comparisons with the Earth Movers' Distance (EMD). The EMD is a valuable alternative to the standard metrics because it considers the complete distributions of vegetation distributions and can assign specific weights to different

types of mismatches. Since the EMD integrates more information, EMD-based studies allow for more refined interpretations, as illustrated through a series of case studies based on biome estimates from pollen samples and simulations. The versatility of the EMD enables performing various types of data-data and data-model comparisons with biome data (as presented here) and with other palaeoenvironmental, palaeoclimatic or archaeological proxies. To complement the use of the EMD, we propose a statistical framework to test the robustness of comparisons (*i.e.* testing if the different elements being compared share similar

features). Finally, the EMD and the EMD-related significance tests have been integrated into an R package *paleotools*, to facilitate access and reuse.

**Code and data availability**. The 'paleotools' R package is available from https://github.com/mchevalier2/paleotools.

**Financial support.** This research has been supported by the PalMod Initiative, funded by the German Federal Ministry of Education and Research (BMBF), Research for Sustainability initiative (FONA, http://www.fona.de, last access: 6th June 2022). In particular, MC is funded by PalMod subproject 01LP1926D, AD by subproject 01LP1920A, JPB and NW by subproject 01LP1926C and UH by subproject 01LP1510C. This research has been supported by the European Research Council (ERC Glacial Legacy 772852 to UH). CL holds a scholarship from the Chinese Scholarship Council (grant no.

201908130165). NW acknowledges funding by the Deutsche Forschungsgemeinschaft (DFG, German Research Foundation), project no. 395588486. XC acknowledges funding from the Basic Science Center for Tibetan Plateau Earth System (BSCTPES, NSFC project No. 41988101) and the Sino-German Mobility programme (grant no. M-0359).

**Author contribution.** MC designed the study, performed the experiments, and wrote the original manuscript. All authors

contributed ideas from the earliest stages and commented on the different iterations of the manuscript.

**Acknowledgements.** MC thanks the Max Planck Institute für Meteorology (MPI-M) in Hamburg and the Climate Vegetation Dynamics group, in particular.

# 8 Appendix 1

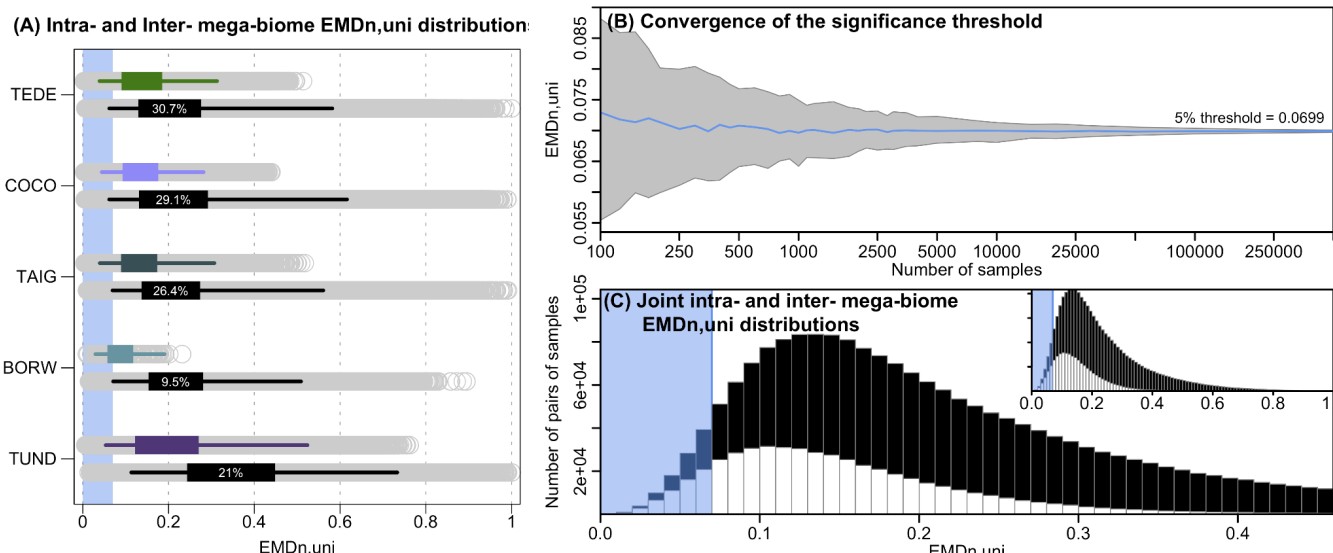

**Figure A1. Distribution of intra- (coloured) and inter- (black) mega-biome EMD$_{n,uni}$ distributions across the study area. In all five panels, the top/coloured boxplot represents the distribution of the pairwise distances of all the samples with the same mega-biome with the highest affinity score, and the bottom/black boxplot represents the EMD$_{n,uni}$ distributions of these samples with different mega-biomes with the highest affinity score. The box of the boxplot represents the 25-75% interval (interquartile range), and the whiskers represent the 2.5-97.5% interval. The percentages indicate the proportion of samples where the EMD$_{n,uni}$ of the inter-biome distribution is lower (estimated from 10,000 bootstrapped pairs of samples drawn from the intra- and inter-mega-biome EMD$_{n,uni}$ distributions). The higher the percentage, the higher the overlap of the two distributions.**

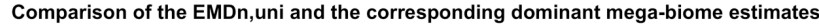

**Figure A2.** Comparison of the $EMD_{n,uni}$ and the corresponding mega-biomes with the highest affinity scores (A–C) in time and (D–E) in space. (A) 'mega-biome with the highest affinity score' reconstruction for a pollen record from northern Italy (Cao et al., 2019; Finsinger et al., 2011; Finsinger and Tinner, 2006). (B) $EMD_{n,uni}$ calculated between contiguous pairs of samples, highlighting that vegetation changes that trigger a change in the mega-biome with the highest affinity score are not different from the changes that do not. (C) $EMD_{n,uni}$ of the biome scores compared to the top sample, highlighting significant vegetation changes across time. The significance threshold at 5% (blue band) was derived from the random sampling of 2000 pairs of Holocene samples across Europe. (D–E) Mapping of the $EMD_{n,uni}$ of all the regional samples compared to the mega-biome reconstruction at the location indicated with a red diamond at 0 BP (D) and 6000 BP (E).

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
