# Peer review of "Refining data-data and data-model vegetation comparisons using the"

_EGUsphere, 2022_

## Author Comment (AC1)

The comments of Reviewer 1 are in black, and our responses are in blue.

Reviewer 1 provided a comprehensive evaluation of our manuscript that contributed to improving its clarity. A large fraction of Reviewer 1's disagreement comes from our choice of using biomes instead of PFTs. Deciding which is best is beyond the scope of the paper, and we have adapted the introduction to reflect that either is a valuable candidate. We now clearly state that we chose biome estimates because biome data is publicly available for more regions and time intervals. The data is also simpler to interpret and, thus, more adapted to illustrate the advantages of the EMD. It has nothing to do with advertising for using biomes for such studies.

The second major criticism was about our penalty matrix. While we acknowledged that the one presented here was again just an example designed to illustrate the power of the EMD, Reviewer 1 wanted it to be more complex. Reviewer 1 offered a credible alternative combining vegetation structural changes (similar to what we used) with changes in climatic zones. Therefore, we implemented it instead of our approach (see revised Sect 2.2.2 and Fig. 4 in the manuscript).

We believe most of Reviewer 1's comments are due to miscomprehension. Therefore, we have reworked the text where we believed clarifications were necessary (see details below). We hope the revised manuscript clarifies the numerous points of contention highlighted by Reviewer 1 and that Reviewer 1 now better perceives the value of our work.

**General comments**

This paper addresses an important problem, that of the inherent limitations of reducing multivariate palynological data to a limited number of biomes, and then using the latter to make univariate comparisons between the biomes inferred from the palynological data and the biomes simulated by vegetation models (i.e. data–model comparisons), or between two sets of biome inferences (i.e. data–data comparisons). Whilst the approach that the authors present does appear to be new, it is disappointing that they fail, in either the introduction or discussion sections of their paper, to cite examples of others who have addressed this issue previously, and who have developed and applied different approaches to overcoming the limitations of the univariate, or binary, comparison approach.

We accept that our review of the field should have been more thorough, and we believe the revised version of the manuscript is now more comprehensive, thanks to the suggestions of Reviewer 1.

For example, Huntley *et al*. (2003, p. 198) wrote: "*Our initial results, however, exposed the principal weaknesses of such biome-based data–model comparisons. First, both the modeling approach and the biomization technique for pollen data involve classifying what are essentially continuously variable data into a limited number of arbitrary categories, i.e., the biomes. Furthermore, these categories are based upon the present vegetation and thus do not accommodate the possibility that vegetation units may have existed in the past, under environmental conditions without any extensive modern analog, that do not match directly to any contemporary biome. Second, biome mismatches between data and model may arise for a number of reasons that cannot easily be deduced simply by observing the mismatch. In addition, because categories are used, the relative degree of mismatches cannot be assessed.*" Huntley *et al*. (2003) then proceeded to develop and apply an approach that compared the scores inferred from the palynological data for a series of plant functional types (PFTs) with the net primary productivity (NPP) of these PFTs simulated by the vegetation model. Although they also made comparisons using the inferred/simulated biomes, those using the PFT scores/NPP values were more informative, enabling interpretation of the mismatches observed when biomes were compared.

We fully agree with that statement and are aware of the limitations of the biomisation methods. Reviewer 1 seems to believe we advocate using biomes as the only tool to compare data and models in the vegetation space. This is not our intention. Our goal was instead to pick a well-known, simple and widely available type of data to illustrate the use of the EMD in performing a range of analyses. Furthermore, biomes are still a common method to aggregate vegetation reconstructions. Syntheses of biome reconstructions are available on continental or global scales and are commonly used to evaluate

palaeoclimate and palaeovegetation model simulations. Therefore, we see a need to improve the evaluation metrics for these types of studies. The EMD method can partially overcome some limitations of the biome method, but it is not limited to biomes. To further stress this, we have reframed the paper to first talk about generic data-data and data-model vegetation comparison and then to focus on biomes as an illustration. This removes the unnecessary focus given to biomised data. These changes impacted the abstract, introduction and discussion in the revised manuscript. However, the core of the manuscript remains focused on our biomised data as an illustration of the abilities of the EMD.

In another example, Allen *et al*. (2010) compared spectra of "relative biome scores" for 11 biomes, derived on the one hand from palynological data, using a modification of the Prentice *et al*. (1996) biomisation approach, and on the other from the annual NPP values simulated using LPJ-GUESS for a set of 47 PFTs that were then assigned to the appropriate 'pollen PFT'. The spectra of relative biome scores were illustrated in Figures A3.1 and A3.2 of their Supplementary Material (https://ars.els-cdn.com/content/image/1-s2.0-S0277379110001769-mmc1.doc). A canonical correlation analysis was then used to investigate the extent to which the spatio-temporal patterns of relative biome scores from the playnological data and the model were similar, revealing strong canonical correlations between the two sets of relative biome scores, leading to the conclusion that: "*The strong canonical correlation between the two datasets supports the use of the LPJ-GUESS results to infer and discuss the structure and composition of last glacial vegetation, as well as the vegetation patterns across the study region during that time.*" (Allen *et al*., 2010, p. 2608).

We were unaware of this study based on canonical correlation analysis, which is a good way of quantifying the similarity between two datasets. However, it would probably fail to assign a statistical significance to the results because it cannot account for the uncertainty of the data – i.e. what would be the explained variance if the data were random? In a perfect world, it would be 0. But this is never the case, especially with data like biomes, where a few vegetation units are much more dominant than others and typically strong spatial- and temporal autocorrelations are present. How "good" 48% of explained variance is remains unknown because the chances of "correct matches" under random conditions should be high.

In addition, we still need to see how canonical correlation analyses could be employed to compare two samples, which is the analytical level addressed by our study. As such, this canonical correlation approach is a good step towards an in-depth assessment of the correlation between data and model but probably not a panacea when comparing individual samples. It is related to our study but not entirely comparable either. We nevertheless referenced this study in the introduction to provide a broader overview of existing metrics and/or approaches.

*"The approach of summarising an array of affinity scores by its PFT or biome with the highest score is thus insufficiently sensitive, and it can lead to a loss of accuracy when comparing datasets (contrasting samples can be assigned to the same category, while similar samples can be assigned to different categories; Fig. 1). Employing continuous metrics that consider the entire affinity score distributions (as opposed to binary assessments of their PFT or biome with the highest score) can thus refine the quality of data-data and data-model comparisons. Many distances commonly used to compare pollen data – such as the Manhattan/Euclidean distance (i.e. calculating the absolute/squared differences between the scores of the same biomes) or the squared-chord distance (e.g. Overpeck et al., 1985) – could be used to measure the dissimilarity of two affinity score distributions. Biome scores have also been directly compared to net primary productivity produced by LPJ-GUESS using canonical correlation analyses (Allen et al., 2010). However, these metrics give the same importance to all the differences without accounting for the fact that all vegetation changes are not ecologically equivalent."*

More importantly, I don't feel that the approach that the authors have adopted is the most appropriate or powerful. The first step in inferring biomes, whether from palynological data or from the current generation of dynamic global vegetation models (DGVMs), is to compute a 'score' for each PFT. These 'scores' are derived either from the percentage values for pollen taxa assigned to this PFT, or from the

NPP, or some other quantity (e.g. carbon mass or leaf area index (LAI)), simulated for the PFT by the model. It therefore makes far more sense to me, whether making data–model or data–data comparisons, to calculate the (dis-)similarities in terms of the spectrum of 'scores' for PFTs, rather than calculating and using biome scores to derive a distance measure, as the authors are proposing. Although better suited to data–model comparisons where the vegetation is simulated using a DGVM, this approach could in principle also be used with BIOME4, the model which the authors use to illustrate their approach, as this model also uses a series of PFTs and calculates their NPP and LAI.

The authors must, at the very least, justify their use of the 'spectrum' of biome scores, as opposed to the PFT scores, because their approach seems to this reviewer to be perverse.

PFT scores might be a more intuitive way to compare models and data, given that this avoids the PFT-to-Biome step. We now more directly explain why we chose biome affinity scores and highlight that this decision is purely driven by our will to illustrate the EMD with simple, widely-available data and concepts. The exact same analyses could have been done using PFT scores. We rather opted for biome affinity scores because that's the data we had access to, and they were sufficiently detailed to reach our objectives with this study. However, we acknowledge this should have been more directly addressed in the introduction, which is now the case. But we also want to highlight that biomes are still commonly used to evaluate simulations, and the data are much more commonly available (e.g. Binney et al. 2017 and Cao et al. 2019 for Eurasia during the last 40 kyr, Prentice et al. 2000 for the Northern Hemisphere and Africa during the mid-Holocene and Last Glacial Maximum, and Dowsett et al. 2016 for the mid-Pliocene).

*"Then, using a series of illustrative case studies based on the already-published biomised data and simulations of Cao et al. (2019), we show how the EMD can perform ecologically-informed comparisons in the vegetation space. While more and more quantitative reconstructions of PFT distributions at regional scales have been published in recent years (e.g. the REVEALS-based studies by Githumbi et al. (2022) or Marquer et al. (2017)), we preferred using biomised data because biomes are currently the most-widespread format of publicly available continental-to-global scale syntheses of past vegetation changes (e.g. Binney et al. (2017) and Cao et al. (2019) for Eurasia during the last 40kyr, Prentice et al. (2000) for the Northern Hemisphere and Africa, and Marchant et al. (2009) in South America studying the mid-Holocene and Last Glacial Maximum, or Dowsett et al. (2016) for the mid-Pliocene). They also have a lower dimensionality than PFT data and provide, as such, a simpler context to explore the advantages of the EMD. Despite our focus on biomised data, it is important to stress that other categorical vegetation formats, such as pollen-based quantitative reconstructions as computed by REVEALS (Sugita, 2007) and Earth System Models, PFT affinity scores (e.g. Huntley et al., 2003; Allen et al., 2010; Henrot et al., 2017), or even the comparison of pollen percentages at the taxa level could have been used for our case studies."*

Huntley, B., Alfano, M.J., Allen, J.R.M., Pollard, D., Tzedakis, P.C., de Beaulieu, J.-L., Grüger, E. & Watts, B. (2003) European vegetation during marine oxygen isotope Stage-3. *Quaternary Research*, **59**, 195- 212.

Allen, J.R.M., Hickler, T., Singarayer, J.S., Sykes, M.T., Valdes, P.J. & Huntley, B. (2010) Last glacial vegetation of northern Eurasia. *Quaternary Science Reviews*, **29**, 2604-2618.

**Specific comments and errors to correct**

1. Page 1, line 19 – Although biome reconstructions have been used in "*data–model comparison studies to understand past vegetation dynamics*", my own experience suggests that this has been done much less frequently than has the use of biome reconstructions in data–model comparisons the aim of which has been to assess the performance of general circulation models (GCMs) and/or, more recently, Earth System Models (ESMs) – see e.g. many papers arising from the Palaeoclimate Modelling Inter- comparison Project (PMIP – see https://hal.archives-ouvertes.fr/hal-03460457/document). I thus feel that the emphasis given here by the authors to using biome

reconstructions in data–model comparisons to improve understanding of "*past vegetation dynamics*" is inappropriate.

*We have entirely rewritten the abstract to make our goals and assumptions more straightforward. Please note that we also enlarged the scope by replacing biomes with vegetation.*

2.  Page 1, line 20 – It is unclear to me what is meant by the authors when they refer to a 'dominant biome'. A quick check of a few definitions of 'biome' online confirmed my own view, that a 'biome' is by definition extensive, albeit that its geographical scale is not strictly defined, and displays a degree of spatial heterogeneity related both to environmental heterogeneity (e.g. varying degrees of soil moisture across a landscape) and vegetation regenerative processes (e.g. secondary succession following a stand-destroying disturbance). Thus at any given location in space and time, only one biome is present, whereas the use of the term 'dominant biome' implies that several biomes are present, one of which dominates over the others.

    Although it does become clear to the reader, upon further reading, that what the authors are referring to is the biome that achieves the highest biome affinity score when using the widely used approach of Prentice *et al*. (1996) to infer the biome from palynological data, the term that the authors have chosen to use seems to me to be both inappropriate and misleading. I suggest that they delete 'dominant' throughout, and refer simply to 'biome(s)'.

    Prentice, I.C., Guiot, J., Huntley, B., Jolly, D. & Cheddadi, R. (1996) Reconstructing biomes from palaeoecological data: a general method and its application to European pollen data at 0 and 6 ka. *Climate Dynamics*, **12**, 185-194.

*We acknowledge that the term 'dominant biome' is not ideal, and in line with a similar comment from Reviewer 2, we have changed it to "biome with the highest affinity score".*

Page 1, lines 27–28 – Following on from the preceding comment, the first listed advantage of using the EMD, and comparing the pattern of 'scores' (i.e. affinity scores), needs to be re-worded. I suggest something along the lines of: "*1. the affinity scores for all biomes are taken into account, rather than focusing upon the single highest scoring biome*".

*See response to comment #2.*

> *"To overcome this limitation, we propose using the Earth Movers' Distance (EMD) to quantify the mismatch between vegetation distributions available in various formats. Using the EMD circumvents summarising the data into a unary biome or PFT estimate by considering the entire range of biome or PFT affinity scores to calculate a distance between the compared entities."*

3.  Page 2, line 49 – Delete spurious 'usually' (emboldened and underlined for emphasis), i.e. replace "*Biomes are usually estimated from pollen data* **usually** *using a two-step algorithm*" by "*Biomes are usually estimated from pollen data using a two-step algorithm*".

*Corrected.*

4.  Page 2, line 51 – For clarity and ease of reading, delete the third of the four occurrences of 'one' in this line, i.e. replace "*and a second one to assign*" by "*and a second to assign*".

*Corrected.*

5.  Page 2, lines 54–55 – A reference is required to support the statement about the practice of using scaling factors, and specifically to support the suggestion that "*Larix percentages are commonly multiplied by 15...*". This is not a practice that I have come across, and is certainly not one that I would consider to be commonly applied.

This procedure has been used to perform climate quantification to maximise the impact of this taxon with a strong climatic significance (see, for instance, Mauri et al. 2015). Unfortunately, we extrapolated this "trick" to the field of vegetation reconstruction, which was a mistake. This discussion has become, however, irrelevant because the sentence has been removed while reworking the introduction.

Mauri, A., Davis, B.A.S., Collins, P.M., Kaplan, J.O., 2015. The climate of Europe during the Holocene: A gridded pollen-based reconstruction and its multi-proxy evaluation. Quaternary Science Reviews 112, 109–127. https://doi.org/10.1016/j.quascirev.2015.01.013

6. Page 2, lines 55–57 – See my earlier comment about what is generally understood by ecologists and biogeographers when referring to a 'biome'. A biome does not "*dominate the immediate landscape at and around the sampling location*", because a biome is, by definition, extensive and internally spatially and temporally heterogeneous. When inferring from pollen data the biome that is represented, the inference is that the location from which the pollen data are derived falls within such an extensive area occupied at that time by that biome. Once again, the concept of a 'dominant biome' is in my view spurious and inappropriate.

It is also important for the authors to clarify what they mean when they refer to "*the immediate landscape at and around the sampling location*". How extensive is the 'immediate landscape' to which they refer? This is extremely important, because the majority of pollen data used to infer past biomes are derived from the sediments of either lakes or mires of such an extent that the pollen record reflects the vegetation not principally of the immediate surroundings of the sampled location, but of the overall region within which the site sampled falls (see e.g. Jacobson & Bradshaw, 1981; Prentice, 1988).

Jacobson, G.L., Jr. & Bradshaw, R.H.W. (1981) The selection of sites for paleovegetational studies. *Quaternary Research*, **16**, 80-996.

Prentice, I.C. (1988) Records of vegetation in time and space: The principles of pollen analysis. *Vegetation History* (ed. by B. Huntley and T. Webb, Iii), pp. 17-42. Kluwer Academic Publishers, Dordrecht.

Our explanation was probably an oversimplified, rushed definition of what a biome is and how it can be inferred from pollen data. Similarly to comment #6, this sentence has been removed from the reworked introduction, as we opted for a slightly different approach to introducing the topic. We refer Reviewer 1 to the revised introduction to see how these changes were implemented.

8. Page 5, lines 144 *et seq*. – The weighting scheme proposed by the authors takes account of vegetation structure, albeit that it only recognises three categories, but does not take into account climatic zones. Thus, the 'cost' of transforming Evergreen Taiga (TAIG) into Deciduous taiga/Boreal Woodland (BORW) is the same as that of transforming TAIG into Warm Forest (WARF), yet the latter pair fall at least two climatic zones apart and almost certainly will in most cases share far fewer pollen taxa/plant functional types (PFTs) than will the former pair. This, it seems to me, renders this weighting scheme of very limited value.

I suggest that the authors might find it useful to look at the climatic zone and structural type matrices used by Allen *et al*. (2020) for their computation of change matrices, and mapping of changes, when comparing biome patterns simulated for different time slices (see the Supplementary Online Material associated with their paper, p. 26 of jbi13930-sup-0001-Appendix.pdf). The approach of Allen *et al*. (2020) could provide a useful model for the refinement of the present authors' 'cost' weighting scheme. Whilst I recognise that the authors' paper is principally about describing and presenting a new approach, such a refinement of the weighting scheme would render the approach more attractive to many readers.

More fundamentally, because the weighting scheme defines the 'distances', and hence the 'costs'

of moving between biomes, the paper needs to consider in some detail how such a weighting scheme, that is more sophisticated and nuanced than the simplistic illustrative example used by the authors, might be developed.

Allen, J.R.M., Forrest, M., Hickler, T., Singarayer, J.S., Valdes, P.J. & Huntley, B. (2020) Global vegetation patterns of the past 140,000 years. *Journal of Biogeography*, **47**, 2073-2090.

We found the penalty matrix developed by Allen and colleagues very creative and effective. As such, we have adapted it to our more restricted set of mega biomes, as explained in sections 2.2.2 and 3.3. The structural decomposition remains the same (forest, open landscapes, deserts), to which we added 3 climatic levels (boreal, temperate, subtropical/warm temperate), making now 9 possible combinations (see new penalty matrix below). Our number of categories are lower than those of Allen et al. (2020) because we study vegetation at the mesa biome level, which isn't as well resolved. This new matrix does not change the main results of our illustrative examples.

*"We also use two different weighting schemes (the cost of replacing a biome with another one, the "dij" from Eq. 1 ) to illustrate how ecological knowledge can be introduced in such studies (Fig. 2). We use a "uniform" scheme where all the biome changes are given the same weight (EMDuni) and an "ecologically informed" scheme (EMDw), where differences are weighted based on vegetation structural differences (forest vs open landscape vs desert) and climate zone preferences (boreal vs temperate vs warm-temperate / subtropical). This dual definition of biome distance follows the work of Allen et al. (2020), in which each biome is assigned to one vegetation structure and one climate zone. The basal distance between two biomes with the same vegetation structure and climate zone is set to 0.5. Then, each difference in structure or climate zone adds an extra cost of 1 (e.g. moving affinity scores from a forest to a desert makes a cost of 2.5; moving affinity scores from temperate to boreal makes a cost of 1.5). This simple weighting scheme illustrates how an ecologically-informed strategy can refine interpretations compared to the uniform scheme. Different research questions or settings could lead to using schemes with more detailed structural and climatic zone categories (e.g. Allen et al., 2020) or alternative weighting schemes based on, for instance, trait differences (e.g. Sato et al., 2021)."*

[Figure]

New penalty matrix.

9. Page 7, lines 186 *et seq*. – The argument for using the fifth percentile (or any other percentile for that matter) from a large set of EMD values calculated for randomly drawn pairs of sets of biome scores as the basis for establishing a 'threshold' value is completely opaque as far as I am concerned.

The EMD values obtained will be strongly influenced by the (un)evenness of the 'true' representation of different biomes in the dataset being used. A dataset dominated by samples from one biome, with only small numbers of samples representing three other biomes, will result in EMD values that are generally smaller than those for a dataset in which the four biomes are evenly represented amongst the samples. In the first case, thresholding at the 5$^{th}$ percentile will result in rejecting as different many pairwise comparisons where the two samples do in fact represent the same biome.

Reviewer 1 misunderstood the goal of the test. As explicitly stated in the original manuscript, it is not a test to determine if two samples represent the same biome, but rather if two samples' score distributions are structurally similar, which is thus very different from what Reviewer 1 understood/expected. We fully explain in the paragraph introducing the test:

*"Importantly, this test cannot determine if two samples represent the same biome. Because biomes are, by definition, broad vegetation units, two samples can be statistically different and be characteristic of the same biome. In such cases, the statistical difference suggests they likely occupy a different position in that biome's vegetation and/or climate spaces. For instance, vegetation samples taken from the cold and warm ends of the temperature range experienced by the temperate forest biome are likely to be statistically different while still being representative of temperate forests."*

The next sentence then explicitly states what Reviewer 1 raised as a second concern regarding the need for a more or less balanced representation of the dataset:

*"This test can only be used if hundreds of affinity score distributions representative of various environments are available to estimate the randomisation distribution. In addition, this type of threshold is only valid for a given study area and/or research question."*

Additionally, biomes vary in the degree and spatial extent of their internal spatial and temporal heterogeneity, especially as this is reflected by palynological data. Thus, sample pairs representing the same biome, but one that is characterised by spatially and temporally extensive heterogeneity (e.g. taiga that has extensive patches at different stages of recovery from stand-destroying fires) will potentially give larger EMD values than will sample pairs representing different biomes, each of which displays only fine-scale heterogeneity and which share a large proportion of pollen taxa.

The test does not aim to determine if two samples are from the same biome (or any vegetation unit) but if their affinity score distributions can be considered statistically similar. So yes, if the two environments are different, the test will not be significant, and the samples will be labelled as "different", irrespective of which biome they have the highest affinity with. Similarly, two samples from different biomes can be very similar in their composition. There is nothing contradictory about that (cf. Fig. 1 and 5C), and our test addresses this specifically.

Whilst the authors do state that "*this type of threshold is only valid for a given study area and/or research question*", I find it difficult to envisage any study in which such an approach to determining a threshold value will be valid, given that in any set of biomes there will be considerable variation between biomes in their degree of spatial and temporal variability.

The threshold will depend on the data selected for the study. The notion of "similarity" is also strongly scale-dependent. For example, one can imagine two extreme cases, with one study focusing on a small patch of forest in a very restricted region and another study studying European vegetation. The notion of "similar vegetation" will, of course, be very different between the two studies. As such, a specific EMD similarity threshold will be determined by the scale of the study and the research question addressed, i.e. scale-dependent and user/research-dependent.

Surely the appropriate approach to determining such a threshold is to compute EDM values firstly for multiple pairs of (surface) samples randomly drawn from those representing a single biome, and secondly from multiple pairs of samples where the two members of the pair are randomly drawn from those representing different biomes. The distributions of the EDM values for 'like' and 'unlike'

comparisons will then provide a basis for assessing a threshold that can be used to distinguish these two classes of comparison, as well as providing a basis for assessing the uncertainties when making such distinctions. Such an approach was applied by Overpeck *et al*. (1985) and Huntley (1990) to determine threshold values of the chord distance dissimilarity measure when seeking to establish whether or not pairs of samples being compared were analogous (i.e. represented the same vegetation unit or biome).

If this was our goal with the test, this method would *surely* be the solution. But as explained above, our goal was different.

Huntley, B. (1990) Dissimilarity mapping between fossil and contemporary pollen spectra in Europe for the past 13,000 years. *Quaternary Research*, **33**, 360-376.

Overpeck, J.T., Webb, T., III & Prentice, I.C. (1985) Quantitative interpretation of fossil pollen spectra: Dissimilarity coefficients and the method of modern analogs. *Quaternary Research*, **23**, 87-108.

10. Page 10, lines 248–250 – In describing the anomaly approach used, the authors state that 'differences' were calculated between palaeoclimate and pre-industrial climate values, and then 'added' to observations. The climatic variables being used, however, include precipitation, for which it is conventional to use multiplicative, rather than additive, anomalies, thus at least partially overcoming systematic biases in the amounts of precipitation simulated by climate models. Hopefully, the authors have simply failed to make clear that the anomaly procedure used for precipitation differed from that used for temperature; if not, then this, in my view, represents a serious deficiency in the work described (albeit that the primary purpose of the paper is to demonstrate the application of EMD). Incidentally, the authors also need to make clear what type (additive or multiplicative) of anomaly was used for cloud cover.

The data used in this paper are reproductions from a published paper (Cao et al., 2019, mentioned across the entire manuscript). In the "Climate and Vegetation simulation" section, we refer to the appropriate source papers for the readers interested in knowing more about the details. We believe providing too many details about these data would not add anything to the paper. In fact, the intrinsic quality of the data could even be considered irrelevant to the present study, as we only use them to illustrate theoretical concepts in relation to the EMD. We do not interpret the results or draw conclusions about vegetation changes or models' skills, etc., anywhere in the manuscript. We only discuss the skill of the EMD. Therefore, we do not describe the data in more detail in the revised manuscript.

11. Page 14, Figure 6 – The figure caption (lines 340–341) refers to "*(blue dashed lines)*", but I cannot see any such lines on the figure.

The caption has been corrected. We understand the confusion; the blue line had become a blue band during one of the figure iterations.

12. Page 15, lines 349–352 – I feel that the statement made here ("*As opposed to the representation based on dominant mega-biomes that suggests a constant vegetation variability across the record, the EMDn,w trends suggest that vegetation changes were rather slow before ~7,000 BP and since ~2000 BP, and more intense in between.*") is not fully supported by the figure presented and upon which it is apparently based. Figure 6A1 does not indicate "*a constant vegetation variability across the record*", as the authors claim, but rather shows periods of relatively little variation and periods of greater variability, with some relatively rapid changes in the character of the vegetation. In particular, Figure6A1 shows rapid oscillations in the inferred biome between *ca*. 2 and 3 ka BP, with a proportion of samples inferred to represent Tundra (TUND) as opposed to Temperate Deciduous Forest (TEDE). Figure 6A3 reflects essentially the same pattern, with a marked trough in the EDMn,w values relative to the top sample, but with some marked variability that seems, unsurprisingly, to reflect the temporal distribution of the samples for which TUND is inferred. Similarly, there is considerable 'noise' in the EDMn,w values relative to the top sample (Figure

6A3) between 3 and 8 ka BP, corresponding to the inference of Cold/Cool Forest (COCO) for some samples, as opposed to TEDE that is inferred for the majority of the samples. Furthermore, the 'noise' is greatest around 6 ka BP when the 'flickering' in the biome inference is greatest.

It seems that Reviewer 1 might have misunderstood the information that is represented on fig. 6A1, as his/her explanations are a bit confusing. The vertical amplitude of change is not a quantification of the change, as each vertical line, long or small, represents the same information, i.e. one biome being replaced by another one. With that definition, the period between 2 and 3 ka, is only marginally more variable than the rest of the record (see fig below). It just happens to be a period of oscillation between two biomes that were randomly assigned to the top and bottom levels of the figure.

Fig 6A3 shows very different information with much higher short-term variability, i.e. the EMD changes of each sample against the top sample between ~3 to ~6/7 ka compared to the period 2-3 ka. This is coherent with what we see in fig. 6A2, but not what we see on Fig. 6A1. If anything, this period corresponds to the period where the flickering between dominant biomes is the least frequent. This highlights that the EMD can pick up a different form of information from the same input data, which might be used to refine interpretation derived from estimates of the biomes with the highest affinity scores.

Perhaps most importantly, we show that the EMD differences between samples that trigger a change in the biome with the highest affinities are not significantly larger than the differences that do not, such as the changes between 9 and 12 ka. We observe many flickering between TEDE and TUND during that time interval on Fig. 6A1. However, the only significantly different pairs of samples are actually pairs that do not correspond to a change in biomes with the highest affinity score. At the very least, this figure demonstrates that focusing purely on the biome with the highest affinity score can lead to spurious interpretations. Between 9 to 12 ka, vegetation was changing, but it was slowly transitioning from TUND to TEDE. This can be derived by interpreting the flickering observed in fig. 6A1, or by using the EMD to can quantify it (Fig. 6A3). In addition, the EMD indicates two major vegetation changes occurring around 2 and 3 ka (6A3), which corresponds to the flickering between 2 and 3 ka in 6A1. However, between the two transitions, the short-term variability in the EMD differences from the top sample is not higher than in other periods without substantial flickering between biomes with the highest affinity scores.

As to the authors' inferences about the rates of vegetation change, based on Figure 6A3, no fully convincing assessment is possible (by them or this reviewer) without first plotting (smoothed values of) the first derivative of the data illustrated in Figure 6A3, i.e. making a plot that shows the rate of change. Nonetheless, as far as one can tell from Figure 6A3, whilst the variability of the vegetation was greater between 7 and 3 ka BP than it was prior to 7 ka BP, the overall rate of change was greater prior to 7 ka BP, and also since 2 ka BP; this is the opposite of what the authors state, but more consistent with what is shown in Figure 6A1.

The rate of vegetation change cannot be inferred from Fig. 6A3, indeed. And it certainly cannot be inferred from its first derivative either because that would assume that the EMD behaves linearly. Knowing the EMD of two samples from a common reference does not inform about their distance except for the particular case when one of the samples is very similar to the reference. Fig. 6A3 shows the distance relative to the top sample, not the sample-to-sample variability that is discussed in the text. This variability is shown on Fig. 6A2. And the higher spread of the sample-to-sample EMD distance between 3 to ~6/7 ka does suggest that the pollen composition was varying more during that period.

13. Page 15, line 365 – A word or words appears to be missing: "*In such cases, the number of agreeing and is used to measure the accuracy, and...*" just does not make sense.

Corrected. "… the number of agreeing and disagreeing pairings is used to …"

14. Page 17, Table 2 – The kappa values tabulated here are so low that they indicate a lack of **any** significant matches; Monserud (1990) suggests that kappa values <0·4 represent only poor or very

poor matches, and that only kappa values >0·55 indicate good or better than good matches – the maximum value shown in this table is 0·08. Such low values, consistent with the 'accuracy' values shown, led me seriously to question both the authors' results and the original publication. I thus obtained a .pdf of the Cao *et al.* (2019) paper cited by the authors. My reading through of this paper revealed no kappa values; I therefore searched the .pdf, using Acrobat, both for kappa, with no results, and for Cohen, likewise with no results. It is **essential** that the authors accurately cite the source of the data portrayed in this table, thus enabling readers to locate and examine their original source.

Monserud, R.A. (1990) *Methods for comparing global vegetation maps*. Working Paper, WP-90-40. International Institute for Applied Systems Analysis (IIASA), Laxenburg, Austria.

Reviewer 1 identified a very major error in this paper. We cited the wrong Cao et al. (2019) reference. The correct reference is: "*Cao, X., Tian, F., Dallmeyer, A., Herzschuh, U., 2019. Northern Hemisphere biome changes (>30N) since 40 cal ka BP and their driving factors inferred from model-data comparisons. Quaternary Science Reviews 220, 291–309*".

In addition, we agree with Reviewer 1 that the Kappa values are extremely low. However, the meaning of these values will not be discussed in this paper, as they were already presented by Cao and colleagues. We highlight that the EMD can reproduce the same patterns and can identify which values are statistically significant (as opposed to the "expert-based" kappa thresholds that always contain a lot of subjectivity and ignore the structure of the data (e.g. if the same raw data are separated into 20 categories more misclassifications than if they are separated into 3 classes). Our data-model example in the manuscript (Table 2 and Fig. 7) shows that using a predefined expert-based threshold could lead to different interpretations of the same data, as we obtained an EMD value of 0.853, which is significant (comparison at 0 BP with EMDuni), and another value of 0.417, which is not (comparison at 0 BP with EMDw). That is because we account for the structure of the data, while pre-defined, generic thresholds do not.

15.Page 18, line 438 – Once again, something is missing/superfluous here; the clause "*The EMD could be introduced as metric toa to support the definition of analogues,*" does not make sense.

Except for the typo "toa" in the middle of the sentence, the rest of the sentence makes sense. We have expanded on this idea to clarify what we mean by that.

*"Penalty matrices on the level of taxa resolved in pollen records could also be developed to integrate the EMD into pollen-based climate reconstruction algorithms since many of the existing techniques, such as the Modern Analogue Technique (Overpeck et al., 1985), are based on the direct comparison of pollen samples (Chevalier et al., 2020). In addition to measuring their statistical differences as is currently done, an EMD-based definition of pollen analogues would also include the ecology of the taxa so that a well-designed penalty matrix could refine the climate reconstructions."*

16.Page 19, line 452 – ...different types of **disagreements**" or "...different types of **mismatches**", but **NOT** "...different types of **errors**". 'Errors' implies that one biome assignment is correct and the other incorrect, whereas in reality it may often be the case that neither the data nor the model is correct!

Agreed. Replaced by "mismatches".

---

## Author Comment (AC2)

The comments of Reviewer 2 are in black, and our responses are in blue.

We really appreciate the positive feedback provided by Reviewer 2, and the reviewer's suggestions allowed us to clarify some important elements regarding what the EMD can and cannot do. We hope Reviewer 2 will find our responses accordingly adequate.

**Review of "Refining data-data and data-model biome comparisons using the Earth Movers' Distance (EMD)" by Chevalier et al.**

Does the paper address relevant scientific questions within the scope of CP?

*Yes, this contribution looks promising as a new method to analyse pollen data for the past and compare them with vegetation model reconstructions. It is thus directly connected with key aspects of paleoclimate reconstruction and, hence, falls fully within the scope of CP.*

Does the paper present novel concepts, ideas, tools, or data?

*Yes. To my knowledge, the application of the EMD to the analysis of pollen data is completely new. This concept is quite interesting.*

Are substantial conclusions reached?

*Yes. The authors prove the applicability of the EMD for analysing/comparing pollen data/vegetation model reconstructions. In particular, they show that the EMD has the advantage of conserving more information from the original pollen data. Reconstructions look to be more stable through time compared to classical methods based on the biome with highest affinity score, because these latter methods do not offer a continuous measure of vegetation state (i.e., they are discrete classes).*

Are the scientific methods and assumptions valid and clearly outlined?

*Yes, the main method used, the EMD, is well explained and references are provided. Other methods are statistical analyses, which relatively well explained.*

Are the results sufficient to support the interpretations and conclusions?

*Yes.*

Is the description of experiments and calculations sufficiently complete and precise to allow their reproduction by fellow scientists (traceability of results)?

*Yes, probably. The authors have even developed a R package that certainly helps for such a reproduction of their results, as well for analysis of other pollen data.*

Do the authors give proper credit to related work and clearly indicate their own new/original contribution?

*Yes.*

Does the title clearly reflect the contents of the paper?

*Yes.*

Does the abstract provide a concise and complete summary?

*Yes.*

Is the overall presentation well structured and clear?

*Yes, it is very well-organised and generally clear.*

Is the language fluent and precise?

*I am not native English-speaker, but language looks fine to me.*

Are mathematical formulae, symbols, abbreviations, and units correctly defined and used?

*Yes.*

Should any parts of the paper (text, formulae, figures, tables) be clarified, reduced, combined, or eliminated?

The paper has a reasonable length. All parts look necessary. Description of Test 2 (lines 201-216) could be improved. It is difficult to read, especially the method used to establish the second EMD distribution.

We fully agree with Reviewer 2 that the second part of the second test was not as clear as it should have been. We have changed the description to the following, and we believe this makes the test more intelligible.

> *Test 2: Considering the parameters of a study, are the data and the simulation (or modern observations) displaying similar spatial patterns? This second test aims to determine if the mean EMD obtained when comparing a simulated (or observed) vegetation map with a large collection of biome affinity score distributions is smaller than expected when comparing two datasets with different spatial patterns. This test is performed in two steps. First, the data are shuffled (each biome affinity score distribution is randomly assigned to one of the modelled values corresponding to a sample location), and the resulting mean EMD across all locations (i.e. spatial mean) is calculated. This is repeated several times to estimate the distribution of spatial mean EMD values under the assumption that the spatial structure in the data differs from the spatial structure of the simulation (null hypothesis). The 5th percentile of that distribution (any other significance threshold could be used) represents the threshold to reject the null hypothesis (alternative hypothesis: the data and the simulation have similar spatial structures). Then, the uncertainty of the observed EMD value can be estimated by measuring the intra-sample variability. To do so, a second EMD distribution is estimated by bootstrapping, i.e. randomly sampling the same number of biome samples with replacement (some samples are selected many times and others excluded) and calculating the EMD of this bootstrapped dataset with the observed/simulated vegetation map. To determine if the data and the simulation display the same spatial pattern, the 95th percentile of the bootstrapped distribution is compared with the 5th percentile of the distribution of the null hypothesis (one-sided test). If the former is larger than the latter, the null hypothesis is rejected, and the spatial structure of the simulated and reconstructed biomes is considered similar. Efron and Tibshirani (1994) recommend performing at least 200 repetitions to estimate the bootstrapped and null hypothesis distributions. This test is called signif_struct() in paleotools, and its use and interpretation are illustrated in Section 5.*

Are the number and quality of references appropriate?

*Yes.*

Is the amount and quality of supplementary material appropriate?

*Yes.*

**Comments**

In their paper, Chevalier et al. establish a new method based on the EMD to analyse pollen data (change in space and time) or compare them to model vegetation reconstructions. This method is novel, quite interesting and promise to be of broad applicability. The paper is well structured and very

well written. It can be published after very minor revision. I have only a few remarks and suggestions that the authors could consider in their revision:

The concept of biome is integrative, i.e., it is used to represent (within classes) the overall vegetation present at a given location. So, only one biome should exist at a given location. Thus, the words "dominant biome" should be avoided, and replaced by something like "biome with the highest affinity score" (with the pollen data).

We originally opted for "dominant biome" because we wanted to avoid repeating "biome with the highest affinity score" multiple times across the paper. However, since both reviewers find this term inappropriate, we now follow their recommendation and opt for "biome with the highest affinity score".

The authors claim that the biomes are discrete quantities, and for that reason, the methods based on the biome with highest affinity score is presented as less robust than the use of the EMD, which is more continuous. However, with the EMD, the authors use the biome concept and their affinity scores. So, some of the "discontinuities" associated to the discrete definition of biomes still remain, especially when mega-biomes are used as done here. Actually, the EMD method developed here could equivalently be applied using plant functional types (PFTs) and PFT scores rather than biomes/biome scores. For instance, Henrot et al. (2017) (Palaeogeogr . Palaeoclim. Palaeoecol. 467, 95-119, 2017) compared model reconstructions with vegetation data at the level of PFT, i.e., using PFT scores. This allows to keep more information from the original pollen data, that are provided at the genus level. In this case, biome maps are just created to illustrate/capture vegetation distribution in a single map.

We believe Reviewer 2 conflated two ideas here: the description of biomes (or mega biomes) as categorical data with the continuity of the metric used to compare them. Usual comparisons assess whether the biomes with the highest affinity scores are the same or not; hence the result is 1 or 0 (*i.e.* discrete). In contrast, comparing distributions of biome scores with the EMD leads to a real number (*i.e.* continuous). The EMD does not aim to fix the problems associated with the categorical data themselves.

We agree that other types of data could be used for the comparison. As discussed in our response to Reviewer 1, we selected biomes because they are simple data that – we believe – are great for illustrating the concepts we are presenting regarding how to use the EMD. Our goal is not to say that biomes are the best way to compare data and models; they probably are not, at least not for all types of applications. While already present in the discussion, we have tried to make this point clearer earlier in the manuscript.

*Then, using a series of illustrative case studies based on the already-published biomised data and simulations of Cao et al. (2019), we show how the EMD can perform ecologically-informed comparisons in the vegetation space. While more and more quantitative reconstructions of PFT distributions at regional scales have been published in recent years (e.g. the REVEALS-based studies by Githumbi et al. (2022) or Marquer et al. (2017)), we preferred using biomised data because biomes are currently the most-widespread format of publicly available continental-to-global scale syntheses of past vegetation changes  (e.g. Binney et al. (2017) and Cao et al. (2019) for Eurasia during the last 40kyr, Prentice et al. (2000) for the Northern Hemisphere and Africa, and Marchant et al. (2009) in South America studying the mid-Holocene and Last Glacial Maximum, or Dowsett et al. (2016) for the mid-Pliocene). They also have a lower dimensionality than PFT data and provide, as such, a simpler context to explore the advantages of the EMD. Despite our focus on biomised data, it is important to stress that other categorical vegetation formats, such as pollen-based quantitative reconstructions as computed by REVEALS (Sugita, 2007) and Earth System Models, PFT affinity scores (e.g. Huntley et al., 2003; Allen et al., 2010; Henrot et al., 2017), or even the comparison of pollen percentages at the taxa level could have been used for our case studies.*

The EMD method allows to measure a (continuous) distance in the multidimensional space phase with the scores of the different biomes. You can thus show for instance (as in Figure 6) how the distance to a present-day biome has varied in the past. However, the biome phase space is multidimensional and

**Commented [1]:** I think simplicity is not really a good argument....I think a better argument would be the availability of continental/global datasets. We have maps of global biome distributions for the PMIP key time-slices, that's why the palaeo-modelling community is using biomes for model evaluation.... And even if the EMD can be used with other kind of vegetation data, the "original" idea was to find a better method to deal with biomes in model-data comparison studies.  I don't think it is the best method to compare PFT distributions. If they are available as quantitative cover fractions, it will work, but not if you just assign the taxa to PFTs. As far as I understood, in the cited study by  Henrot et al, they just compare the occurrence of  different PFTs, i.e. agreement = (number of sites at which model and data show the same PFT + number of sites at which they both don't show the PFT) / total number of sites.....
That is probably not a good example....

the distance does not tell you in which direction you move. Are you moving towards more forests (and if yes towards which type of forests) or towards more grasslands or deserts? This information is quite important to characterize past vegetation. So, the EMD alone is not sufficient to reconstruct precisely past vegetations. It must be combined with a measure of directions in the phase space. In the method presented here, this role is played by the change in the biome with highest affinity score (so the classical method). But could it be improved to achieve a continuous method to evaluate such directions?

It is true that the EMD only measures how dissimilar two samples are and does not say anything about the types of differences. This is similar to the binary evaluations of biomes with the highest affinity score (they only assess if the biomes are the same or not), which is the metric we aim to replace. Both metrics are insufficient to characterise the direction of past vegetation change. Additional analyses, interpretations and/or external knowledge are necessary to characterise the types of changes. We do not foresee any obvious way to include a direction of change into the EMD itself since mathematical metrics by definition do not contain information on the direction of change. Such direction could be determined "a posteriori" by closely looking at the biome affinity score distributions and EMDs, using the biome with the highest affinity score only or the overall amount of affinity scores with forested biomes, for instance.

However, the computation of the EMD could offer some additional insights into the direction of changes through the "optimal flows" that "transport" the affinity scores of sample A to the affinity scores of sample B (see Sect. 2.1.). The optimal flows, which minimise the transport cost, can be written as a transport matrix. As this matrix contains information on how much affinity score needs to minimally be transported to transform the affinity score distribution from sample A to sample B, it does also contain information on the direction of the mismatch between samples. However, quantifying and interpreting this information is not trivial, because (a) the optimal flows are not necessarily unique (in fact, they will rarely be unique in the case of uniform weights since in this case all flows share the same costs), and (b) the form of the transport matrix depends on the weighting scheme. In principle, the directions of the mismatch will be ecologically more meaningful, the more meaningful the weighting scheme is. Incorporating methods to interpret the transport metrics should be explored in future research and the ecological interpretability of transport matrices could be another major advantage of the EMD compared to other metrics that do not contain information on the direction of change. However, it is beyond the scope of this paper where we are not aiming at a metric for the "vegetation phase shift". Here we only introduce a continuous and ecologically-informed metric to replace a binary one. In addition, and as discussed in length in our response to Reviewer 1, we only used biomes as an illustration of the potential of the EMD to compare multivariate datasets. We do not want to "over-specialise" the metric to biomised data, and we prefer to keep it more generic and usable with other datasets, including PFT scores, as suggested above. This element of discussion is nevertheless very interesting, and we have expanded on it in the discussion of the manuscript as well as a discussion on using the optimal flows in the EMD computation to quantify the direction of vegetation changes/mismatches.

> *"As with most distance metrics, the EMD only measures how dissimilar two samples are and does not provide direct information on the type of (multidimensional) direction of differences. For example, the EMD cannot tell whether sample A is more forested than sample B. It can only quantify how different samples A and B are. This is similar to the binary evaluations of biomes with the highest affinity score. While it is common practice to characterise the direction of change by analysing the properties of the compared datasets separately, the computation of the EMD could offer more direct insights through the "optimal flows" that "transport" the affinity score distribution of sample A to the affinity score distribution of sample B (see Sect. 2.1). The optimal flows, which minimise the transport cost, could be written as a transport matrix. Therefore, this transport matrix would contain information on the (multidimensional) direction of the mismatch between samples. However, quantifying and interpreting these flows is challenging because (a) the optimal flows are not necessarily unique (in fact, they will rarely be unique in the case of uniform weights since, in this case, all flows have the same cost), and (b) the form of the transport matrix depends on the penalty matrix and thus the level of ecological complexity implemented in the penalty matrix. As such, the ecological interpretability of transport matrices could be another*

*advantage of the EMD compared to other metrics. Therefore, we believe that methods to interpret the "optimal flows" should be explored in future research."*

*"Finally, it is essential to emphasise that the EMD, as presented in this study, is not limited to vegetation studies. It can be used with any form of discrete palaeodata (i.e. ordinal and categorical) from different disciplines, including but without being limited to, all palaeoecological datasets (e.g. chironomids, foraminifera, rodents, etc.), geochemical datasets (e.g. n-alkane distribution from terrestrial or marine sediments), or archaeological datasets (e.g. lithics and tools from archaeological deposits). More generally, while raw data counts with a different total number of fossils/artefacts cannot be directly compared with the EMD, their percentages always can because they sum to 100. Said differently, any two samples can be compared with the EMD, provided they have the same total mass."*

I have not found many typos. Only on line 436, "toa" does not make sense. I guess the authors want to say: "... as a metric to support ..."

Typo corrected.

The authors may want to discuss topics (2) and (3) in the discussion section.

We absolutely agree. See our responses to topics (2) and (3) above and our improved discussion in the manuscript.

---

## Author Response (AR2)

The comments of the Editor are in black, and our responses are in blue.

We really appreciate the positive feedback provided by the Editor on our revised manuscript. We have implemented most of the suggested changes, and we detail our choices a little more below.

The authors have carried out a very substantial revision of the manuscript. The new version now covers related literature and is much more balanced concerning the advantages and disadvantages of the different approaches and metrics for analysing the differences between data- or model-based reconstructions of, in particular, past vegetation reconstructions. The authors also include new analyses based on the suggestion from one reviewer.

The presented method is novel and will likely be adopted and further developed by colleagues working in the field. I have some further suggestions and questions for improving the manuscript. The necessary revisions would be minor.

Major comment:

It would be nice to finish the abstract with a sentence on what has been learned by applying EMD to the previously published study on past simulated and reconstructed biomes for Europe. Just stating that the method has been applied does not demonstrate any new insights. The scientific/knowledge advance of the example application should also be very clear in the discussion.

We have included another sentence to highlight some of the benefits of using the EMD, namely that 1) the EMD allows to reconstruct more gradual biome changes while unary estimates look more flickering and 2) we were able to determine which comparisons were statistically significant from those that were not, while accounting for the multidimensionality of the underlying biome affinity score distributions.

Minor comments and questions:

1.) Is "unary" vegetation estimates really the right term? The term is, at least, not usual, and how the unary numeral system applies to past vegetation reconstructions might not be clear for all readers.

We have looked for other terms since the beginning and settled for "unary" because it was the one that ticked the most boxes. We acknowledge this term is unusual but we believe that our definition of the meaning of this word works in the context of this paper and will enable interested readers to follow.

2.) Abstract: Better "when minor variations in pollen percentages occur" instead of "when minor variations in pollen percentages change"?

We agree that this sentence could have been more clear. However, we replaced "change" by "modify" as we believe it is less ambiguous in this context.

3.) Abstract, "to quantify the mismatch between vegetation distributions available in various

formats": Can it be made clearer what is meant by "vegetation distributions"? Maybe reconstructions of "biome distribution and vegetation composition"? "Available in different formats" sounds like a technicality. Necessary here?

Rephrased as follow: "To overcome this limitation, we propose using the Earth Movers' Distance (EMD) to quantify the mismatch between vegetation distributions (e.g. between distributions of affinity scores)"

4.) Please, check carefully how literature is cited. In the introduction, Allen et al (2020) is cited for determining the global accuracy between data sets, but Allen et al. (2020) mainly addressed the magnitude of past and future simulated biomes shifts, not some kind of accuracy between the model and data, at least not quantitatively. Furthermore, a term like "correspondence" would be more suitable in the sentence than "accuracy".

We replaced accuracy with similarity to better account for the diversity of studies involved in our references. We also split the references to better reflect which reference refers to what.

"(Prentice et al., 1996, 2000). The transformed pollen data can thus be directly compared with model simulations of the same period (e.g. Cao et al., 2019; Prentice et al., 1998) or other pollen data of different periods (e.g. Allen et al., 2020) and the "agreeing" and "disagreeing" pairings (i.e. binary assessments of the compared biome or PFT estimates with the highest affinity score) are counted to determine the global similarity of the compared datasets."

5.) When Allen et al. (2010) is cited the first time, I would make clear that the conical correlation analysis was based on NPP "per PFT" (whereby modelled PFTs were matched with pollen PFTs). This important detail is not clear from the formulation so far. In other words I would add "per PFT" or so.

Yes and no. Based on Allen et al. (2010), the analysis was performed at the biome level (what we wrote). It just happened that the biomes were derived from NPP per PFT. We included that information as follow: "Pollen-based biome scores have also been directly compared to biome scores estimated from the net primary productivity per PFT produced by LPJ-GUESS using canonical correlation analyses (Allen et al., 2010).".